# SFT: Sampling-based Foundational Transformer

## Abstract

The versatility of self-attention mechanism earned transformers great success in almost all data modalities, with limitations on the quadratic complexity and difficulty of training. To apply transformers across different data modalities, practitioners have to make specific clever data-modality-dependent constructions. In this paper, we propose one efficient transformer that can work on multiple data modalities (point cloud, graph, and sequence) and constraints (rotational-invariant) without any modifications. The existence of such model is important as contemporary foundational modeling requires operability on multiple data sources. For efficiency on large number of tokens, our model relies on our context aware sampling-without-replacement mechanism for both linear asymptotic computational complexity and real inference time gain. For efficiency on the training convergence rate and to ease the pain of meticulous hyper-parameter tuning, we rely on our newly discovered pseudoconvex formulation of transformer layer. As a foundational model, we achieved competitive results on many benchmarks, while being faster in inference, compared to other very specialized models. We release our source code in supplementary materials.

## 1 Introduction

Transformers (Vaswani et al., 2017) have been successfully applied to a wide variety of tasks on many data modalities, namely sequence (Vaswani et al., 2017; Radford et al., 2019; Devlin et al., 2019), vision (Dosovitskiy et al., 2021; Carion et al., 2020; Liu et al., 2023), speech (Gulati et al., 2020; Gong et al., 2021; Koutini et al., 2022), time-series (Liu et al., 2022; Jin et al., 2024), point cloud (Lee et al., 2019; Zhao et al., 2021; Yang et al., 2019), and graph (Yun et al., 2019; Rampášek et al., 2022; Shirzad et al., 2023; Ngo et al., 2023). Its successes can be characterized by the versatility and parallelizability of self-attention modules in long-range modeling. This implies transformers can transform the whole sequence simultaneously like convolutional neural networks (CNN) with unlimited receptive field. This alone allows the model to scale up to a very big size, enabling large language modeling. While being versatile and parallelizable, transformers have some limitations: heavy architectural modifications for other data modalities adaptation, inherent quadratic complexity that limits scalability, and certain training difficulties. The first limitation arises when recent developments in deep learning towards foundational modeling. Large foundational models need more data beyond text (e.g. SQL databases, knowledge graphs, etc.) and end-users want to input different data modalities into these models like normal chatting experience between humans (e.g. "Look at my cute dog <dog.png> with her puppies <puppies.png>."). The exceptionally high-quality and abundant (probably in private) knowledge graph data and the relationship between different data modalities is an interesting research direction. The very first step towards such models is designing one single architecture for a large number of data modalities and evaluating its efficacy on each of these data modalities.

In this work, we focus on solving the mentioned limitations with our new relative positional encoding scheme which works (adequately) for many kinds of data modalities. As our research resources have certain constraints, our research limits to two common archetypes of relationships between tokens: **dense low rank** (point cloud) and **sparse high rank** (sequence and graph). Graphs also pose a different challenge: the structural patterns are much more diverse compared to sequences. For training and evaluation efficiency, we lean towards developing sparse global attention, and some tweaks to trivialize the training process. We are interested in sparse global attention because its simplicity allows: applicability on (almost) all data modalities

and transformer variants, a subquadratic asymptotic complexity, and a **lower practical runtime**. We notice that most sparse attention mechanisms either utilize a fixed sparse pattern (Beltagy et al., 2020; Child et al., 2019) or rely on random sparse patterns (Shirzad et al., 2023) or a combination of these (Zaheer et al., 2020). Our initial idea is to use a differentiable sampling method to learn a subsampling pattern. Based on Gumbel-softmax reparameterization, we design sampling **without** replacement through neural importance score computation. This way, the sparse global self-attention can learn to attend to important tokens and is usable on non-sequential data structures like point clouds and graphs. The attention nonlinearity we design is derived from the maxout network(Goodfellow et al., 2013). We combine it with a ReLU-based probability function instead of the Softmax function previously seen in transformers. The combination ultimately makes each of our transformer cells pseudoconvex on its parameters and is linear-scaled. This greatly alleviates the hyperparameter tuning pain of transformers. We further show that our maxout attention nonlinearity combined with the ReLU-based probability function allows better relative information aggregation. To demonstrate the effectiveness of relative information aggregation via positional encoding, we apply it to rotational invariant point cloud tasks, with coordinate information masked out of the input sequence. In this setting, only transformed Euclidean distance is introduced to the attention matrix. This technique (the ReLU-based probability function and the convex attention non-linearity) can be applied to any other self-attention-based model to increase the model expressivity.

In this paper, we conduct extensive experiments on point cloud, graph, and sequence datasets. For point clouds, we benchmarked the model via classification (Wu et al., 2015) and semantic segmentation (Chang et al., 2015; Armeni et al., 2016) tasks, in both conventional paradigms as well as rotational insensitive ones. For graphs, we evaluate our method using two Peptide datasets (Singh et al., 2015) and one Computer Vision dataset (Everingham et al., 2010). For sequences, we used the long-range arena benchmark (Tay et al., 2021) featuring four classification tasks and one retrieval task on these datasets: ListOps (Nangia & Bowman, 2018), IMDb review (Maas et al., 2011), ACL Anthology Network (Radev et al., 2009), Grayscaled CIFAR-10 (Krizhevsky & Hinton, 2009), and Pathfinder (Linsley et al., 2018). We achieve competitive results on these benchmarks as a foundational model working on both point clouds and sequences compared to the specialized ones, featuring faster empirical runtime.

In short, our contributions can be summarized as follows:

- Efficient linear complexity sparse self-attention in terms of the number of tokens,

- Differentiable and parallelizable sampling without replacement method that is optimization-friendly,

- Pseudoconvex transformer layer that is stable during training by theoretical gradient analysis,

- Zero-additional computing cost rotation-invariant addon for point cloud transformers,

- Outperformance over many well-known methods on standard benchmarks.

We provide the detailed theoretical analysis in the Appendix Section B.

## 2 Related works

Our work provides an efficient transformer variant that works on three data modalities: point clouds, graphs, and sequences; therefore, this section provides a context on relevant literature on efficient transformer, point cloud learning, graph learning, and sequential learning literature.

**Efficient Transformers.** The scalability of transformers is hindered by the quadratic complexity of self-attention modules. Hence, numerous works have delved into sparsity (Beltagy et al., 2020; Ainslie et al., 2020; Zaheer et al., 2020; Shirzad et al., 2023), low-rank projection (Wang et al., 2020), hashing-based method (Kitaev et al., 2020), and kernel-based methods (Choromanski et al., 2021; Katharopoulos et al., 2020) to sacrifice the expressivity of transformers for a sub-quadratic construction. Our review of contemporary efficient transformers is influenced by this survey by (Tay et al., 2022). Linformer (Wang et al., 2020) is based on the low-rank assumption of self-attention matrices, realized via projection matrices. As one of the earliest

works in efficient transformers, it has flaws in both scalability and fixed sequence length. Subsequent work like Linear Transformer (Katharopoulos et al., 2020) redesigns the similarity kernel previously as a softmax of dot-product score (or any other kernels) to one made of a linear probability cage. The modification frees them from the recomputation of the multiplication of key-value matrices, making attention computational cost $O(N)$, where $N$ refers to the number of tokens. Methods like (Beltagy et al., 2020) utilize three choices of fixed-sparsity-patterns: sliding window, dilated sliding window, and global+sliding window. Similarly, (Zaheer et al., 2020) uses a combination of different sparse patterns at one time: random attention, window attention, and global attention. On graphs, (Shirzad et al., 2023) also combines different sparsity patterns but with a constraint on the random-generated one: it has to be an expander graph. The hard-coded sparsity effectively linearized the complexity of transformers and is empirically fast. However, intuitively, no pattern or finite set of patterns fits all problems; therefore, there is also a line of work dedicated to a differentiable sparsity. (Roy et al., 2021) drew a relationship from MIPS problem (Maximum Inner Product Search) and the Nearest-Neighbor Search algorithm, when both the query and key vectors are unit vectors. With their online k-means algorithm, their method attends each query vector to every key vector belonging to the same cluster, thus, bringing the complexity down to $O(n^{1.5})$, with $n$ is the number of tokens. Empirically, however, their method is not fast as it needs a k-means construction and extensive specialized implementation for sparse operators. We see that the literature currently lacks a fast learnable sparsity and this is what we provide, based on sampling method. (Lee et al., 2019), is most similar to our global attention scheme, as it is also based on sampling. Their method uses a multi-head attention stacked on one another to perform sampling. Except, our method learns a sparsity pattern for global attention, which has been proven to be more robust.

**Point Clouds.** Deep Learning has been applied in Point-Cloud-related problems, including 3-D Shape Classification, 3-D Point Cloud Segmentation, and others. Our literature review on Point Cloud methods is inspired from (Guo et al., 2021; Zhang et al., 2023; Gerken et al., 2023) and other contemporary literature, covering a wide range of geometric deep learning. Data structure-wise, point cloud works can be classified into three types: point/mesh, voxel, and multi-view; applied by different deep learning methods. Multi-view based methods like (Su et al., 2015; Yu et al., 2018) process point cloud via a sequence of rendered 2-D images of objects from different views, enjoying the early breakthroughs of deep learning-based computer vision (He et al., 2016). The voxel-based method (Maturana & Scherer, 2015; Qi et al., 2016) operates on a 3-D grid representing the object, which also uses convolutional neural networks. The voxel-based method is computing-intensive: the number of voxels grows cubically to the grid resolution. Therefore, numerous methods like (Riegler et al., 2017) are dedicated to the sparsification of the voxel grid. OctNet (Riegler et al., 2017) sparsify the voxel grid using an octree to allocate more memory and computation resources to denser areas of the voxel grid. In contrast, point/mesh-based methods like the works by (Charles et al., 2017; Qi et al., 2017; Zhao et al., 2021) use unprocessed inputs (3-D coordinates with/without surface normal vectors). These works often share several components: local permutation invariant point operators, global feature aggregation, and point downsampling operators. Methods-wise, point cloud works can be classified into types: local-aggregation-based methods and global-aggregation-based methods. Local aggregators in point cloud literature usually feature convolutional layers or point-to-point global aggregators with self-attention. Convolutional-powered local aggregators can take many forms depending on the data structure: 2-D convolutional layers (Su et al., 2015; Yu et al., 2018) for multi-view, 3-D convolutional layers (Maturana & Scherer, 2015; Qi et al., 2016) for voxel grid, graph convolutional layers (Shi & Rajkumar, 2020) for point graph (usually constructed using kNN or distance ball), and kNN-powered local feature aggregator (or point convolution operators) (Qi et al., 2017; Xu et al., 2018). Point-Conv (Wu et al., 2019) and Point-Transformer (Zhao et al., 2021) also feature point interpolation operators as an aggregation method. Global feature aggregation in point cloud literature is dominated by the likes of self-attention (Vaswani et al., 2017) popularized by the fact that the module has been so successful in modeling long-range dependencies in sequential processing. Point-Transformer (Zhao et al., 2021) is one of the works featuring long-range dependency modeling. It also takes account of each point's locality via nearest neighbor max pooling, which also allows the model to operate on higher-level features as they progress through deeper layers.

**SO(3) Point Clouds.** As point clouds exist in a 3-D Euclidean space, the rotation of objects should not change what those objects are. This is called the rotational invariance characteristics of some point cloud models and is an active area of research. Early research relies on data augmentation to achieve rotational invariance; however, it should be noted that covering all possible rotations is not achievable. Subsequent

ones rely on modules satisfying one of the two characteristics: rotational invariant (Sun et al., 2019; Li et al., 2022; 2023) or rotational equivariant (Chen et al., 2021; Anderson et al., 2019; Deng et al., 2021; Kaba et al., 2023). Equivariant-based methods in point cloud, as a generalization to invariant-based ones, feature diverse mechanisms satisfying the equivariant constraint: $f(r(x, \alpha)) = r(f(x), \alpha)$, where $f$, $r$, and $\alpha$ is the equivariant mapping, the rotation function, and the rotation angle, respectively. As an early work, similar to the existing equivariant literature in other data structures like images, (Chen et al., 2021) develops a spherical point convolution operation. (Anderson et al., 2019; Thomas et al., 2018; Cohen et al., 2018) propose learning on Fourier domain by utilizing spherical tensors and Fast Fourier Transform on $SO(3)$ group. This allows them to retain perfect information of the whole point cloud on each layer, sequentially without a complicated multi-path model structure. (Deng et al., 2021) extends each neural node to have 3-D representation, with different linear projection and non-linearity. Their method is elegant and simple enough to apply to any other point cloud framework. It sees performance degradation in (Charles et al., 2017) but retains full performance on (Su et al., 2015). In rotational-invariant literature, Euclidean distance is the most commonly used feature (Jing et al., 2021; Maron et al., 2019; Li et al., 2023). The reason: it is simple and it contains all geometric information of any point cloud (Satorras et al., 2021). (Li et al., 2023) further shows that GNN is incapable of distinguishing certain point cloud cases given distance matrix and proves that a k-tuple-distance is capable of capturing any k-order geometric features. In line with other rotational-invariant via GNN methods, we show that rotational invariance-via Euclidean features can be efficiently applied to transformers in the form of relative positional encodings, with significant architectural changes. In addition, we provide an empirical analysis of the ability of our transformer on performance and efficiency on popular point cloud, sequence, and graph benchmarks with relative positional encodings.

**Graphs.** Graphs are used to represent relative data with a variety of use cases, e.g., small atomic structures (Wu et al., 2018; Townshend et al., 2022) for drug discovery, knowledge graphs (Hogan et al., 2021; Ko et al., 2021; Choudhary et al., 2021) for reliable information retrieval, etc. Modern graph pattern recognition relies on deep learning with the emergence of Messaging Passing Neural Network (MPNN) (Gilmer et al., 2017). It is a learning paradigm where the computation of each node feature relies solely on its locality. Graph Convolutional Network (GCN) (Scarselli et al., 2009; Kipf & Welling, 2017) is the earliest work of this kind, where each node aggregates the feature of adjacency nodes using a permutation invariant operator. (Hy et al., 2018; 2019) extends the GCN model to higher-order while respecting the permutation symmetry by proposing the use of covariant tensor operators. Graph Attention Network (GAT) (Veličković et al., 2018) improves GCN by introducing a mechanism to select important nodes. These two are the classical graph neural networks and they suffer from overfocusing on locality; which prevents them from drawing meaningful long-range relationships (Dwivedi et al., 2022b; Ngo et al., 2023; Trang et al., 2024). Researchers have designed two mechanisms to tackle this with MPNN: virtual nodes (Pham et al., 2017) and k-hop neighborhood (Nikolentzos et al., 2020). Virtual nodes are designed to be nodes that connect to all nodes. This design provides information shortcuts for any two distant nodes, allowing global information to be propagated throughout all graph nodes. This concept is later explored and proved to be equivalent to low-rank-approximation-based transformer networks (Cai et al., 2023). K-hop MPNN aggregates information not only from the adjacency nodes but also the set of nodes that can be reached from the central node by following a path with a maximum of k edges. This increases the local receptive field of each graph node, which alleviates the mentioned problem, similar to the large convolutional kernel trend in computer vision (Ding et al., 2022). Transformer(Vaswani et al., 2017) operating on the complete graph domain (Bronstein et al., 2021), overcomes the mentioned problem purely through architectural design. While transformers are useful in extremely-large-scale learning tasks like language modeling, their performance is no good compared to traditional graph transformers in graph datasets, which often have few graphs with large numbers of nodes. Therefore, a considerable body of literature is devoted to making better handcrafted encodings as graph inductive bias (Ying et al., 2021; Dwivedi et al., 2022a; Rampášek et al., 2022). Graphormer (Ying et al., 2021) is one of the first graph transformer. It introduces several types of encodings: centrality encoding (how important the nodes are) and spatial encoding (how important the locality of each node is), which resembles how humans infer useful information from graphs. GraphGPS (Rampášek et al., 2022) designs a general pipeline for graph modeling by combining many prominent techniques into one: positional encoding (local PE, global PE, and relative PE), structural encoding, and multiple graph block types (MPNN layers (Brossard et al., 2021; Bresson & Laurent, 2018), attention layers (Vaswani et al., 2017; Choromanski et al., 2021;

Zaheer et al., 2020)). Furthermore, (Ngo et al., 2023) proposes a multiscale version of graph transformer that learns to construct a hierarchy of coarsening graphs and a new form of positional encoding via graph wavelet transform. Our work focuses more on the scalability of the quadratic complexity graph transformer and the representative power of our relative information aggregation. This problem has been recently addressed by (Shirzad et al., 2023) where the attention mechanism is sparsified by virtual node and expander graphs. Learnable sparsity for sub-quadratic graph transformer is a relatively new technique, recently explored by (Trang et al., 2024). While the idea seems new and intriguing, it suffers from low convergence rate from stacked Gumbel-Softmax sampling layers to smoothen the discrete hierarchy. Here, we designed a more optimization-friendly differentiable sampler that can be stacked onto each other and a much simpler sparsity pattern (sampling versus hierarchy).

## 3 Background

Variables (scalar, vector, matrix, and tensor) are all implemented as **tensors**. In practice, we use lots of tensor operators, namely, matrix multiplication, tensor concatenation, tensor stack, common activation functions (ReLU, Softplus, exp), slicing, sorting along dimension, gather, transpose, and permute. These mentioned operators should be explained in common deep learning frameworks' documentation, namely, PyTorch, TensorFlow, ... Stating this, however, we try to limit the usage of tensors in our mathematical equations and resort to scalars, vectors, and matrices whenever possible. In all of our equations, there are two occasions we used Stack function: to define the sampling operator and to illustrate the learnable sampling with replacement; therefore, we provide the definition of Stack function here. The Stack function receives $n$ $d$-dimensional vectors and returns a matrix having shape $n \times d$. The elements of the resulting matrix are expressed as follows:

$$\text{Stack}(\mathbf{v_1}, \mathbf{v_2}, \ldots, \mathbf{v_n})[i,j] = \mathbf{v_i}[j] \tag{1}$$

The self-attention mechanism is a core component of transformer models to capture long-range dependencies. Let $\mathbf{X} = \{\mathbf{x}_1, \mathbf{x}_2, \ldots, \mathbf{x}_n\} \in \mathbb{R}^{n \times d}$ represent a sequence of $n$ $d$-dimensional vectors, $\mathbf{Q}$, $\mathbf{K}$, and $\mathbf{V}$ vectors representing query, key, and value vectors are created using three learnable linear transformation: $\text{Linear}(\mathbf{X}) : \mathbb{R}^d \to \mathbb{R}^d$. Then, using the $\mathbf{Q}$, $\mathbf{K}$, and $\mathbf{V}$ vectors, it creates another sequence by aggregating tokens satisfying certain criteria, expressed as follows:

$$\text{Attn}(\mathbf{X}) = \text{Prob}\left(\text{Score}(\mathbf{Q}, \mathbf{K})\right) \mathbf{V}, \tag{2}$$

where Prob and Score are attention non-linearity and attention score functions, respectively. In practice, $\mathbf{Q}, \mathbf{K}, \mathbf{V}$ are all tensors with the dimension of batch and attention heads; however, it is processed the same way concurrently across all batch elements and attention heads. Therefore, it is safe to formulate as one attention head, where $\mathbf{Q}, \mathbf{K}, \mathbf{V}$, and $\text{Prob}(\text{Score}(\mathbf{Q}, \mathbf{K}))$ as matrices.

$\mathbf{Q}, \mathbf{K}, \mathbf{V}$ are matrices of size $n \times d$. $n, d$ in the following equations are the number of tokens and the number of token dimensions, respectively. The function $\text{Prob} : \mathbb{R}^{n \times n} \to \mathbb{R}^{n \times n}$ and $\text{Score} : \mathbb{R}^{n \times d} \times \mathbb{R}^{n \times d} \to \mathbb{R}^{n \times n}$ can be any matrix(ces) to matrix transformation (linear or non-linear). This definition provides a generalization on what a transformer network can be. The earliest transformer (Vaswani et al., 2017) uses softmax and scaled dot product functions as Prob and Score. Replacements of Prob or Score compared to the vanilla transformer (Katharopoulos et al., 2020; So et al., 2021; Wortsman et al., 2023; Shen et al., 2023) are created to address the problem of quadratic complexity, over-centralized attention, and ease of training or just motivated purely from empirical results. In our work, we introduce changes to both Prob and Score to take care of the mentioned problems and further improve the theoretical expressivity of transformers, regarding relative positional encodings.

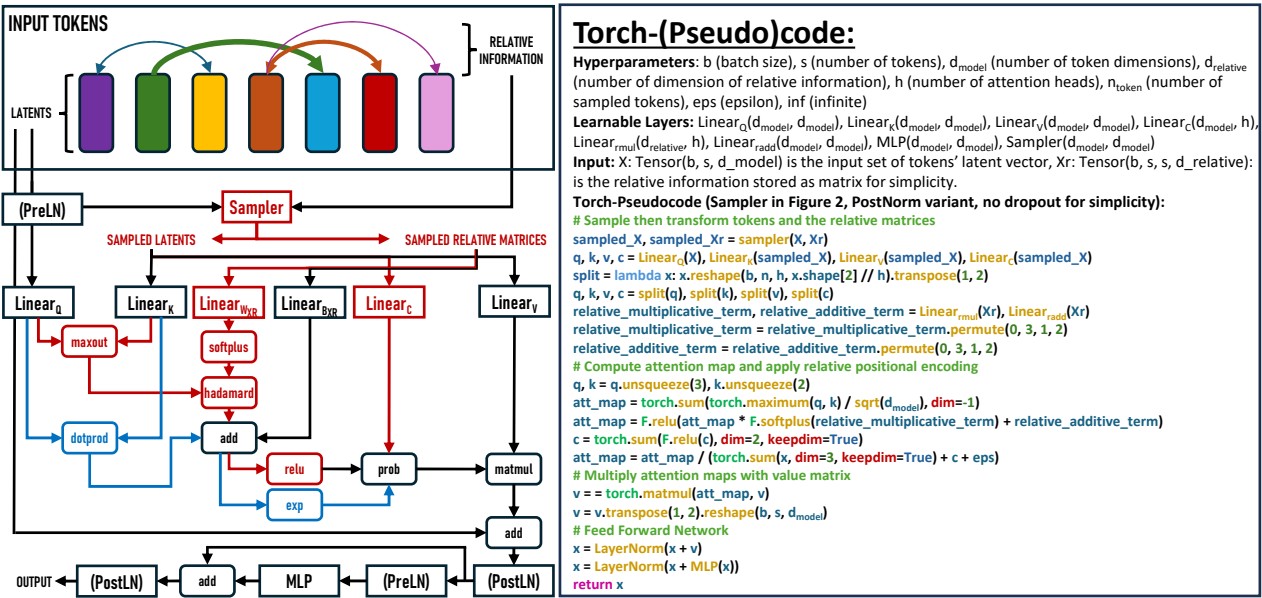

Figure 1: Our proposed sampling-based transformer variant. (PreLN) indicates LayerNorm if Pre-LN Transformer variant, otherwise, identity. (PostLN) indicates LayerNorm if Post-LN Transformer variant, otherwise, identity. Red (our model) and blue (vanilla transformer) highlight the difference between our model and vanilla transformer (with attention bias). Sampler is described in Figure 2.

## 4 Method

### 4.1 Overview

Our method is a composition of neural-based sampling without replacement method, modifications to the self-attention mechanism with rigorous theoretical justifications, and supporting layers. Similar to Linformer (Wang et al., 2020) applying low-rank approximation to key and value vectors, we delve into the sampling procedure for key and value vectors, elaborated in Section 4.2. Next, Section 4.3 describes our easy-to-optimize attention score combined with relative positional encodings via leaky probability formulation. Finally, the layer ends with a feed-forward network, transforming the vector representation of the tokens.

The method is summarized in Figure 1.

### 4.2 Differentiable Sampling Without Replacement

We formally define the operator that samples $k$ of $n$ real $m$-dimensional vectors as a function $\text{Sam}(\mathbf{X})$ : $\mathbb{R}^{n \times m} \to \mathbb{R}^{k \times m}$. Sampling is a linear transformation, the resulting sampled vectors can be defined by a matrix multiplication between the selection matrix $\mathbf{P}$ and the input token matrix $\mathbf{X}$: $\mathbf{PX}$. However, not every linear operation is a sampling operation; there are certain properties that $\mathbf{P}$ has to be met, so that, the linear transformation can be a sampling operation. That is: for the linear transformation to be an operator that samples $k$ of $n$ real $m$-dimensional vectors $\mathbf{P}$ must be representable in form of a stack of $k$ one-hot vectors defined as follows:

$$\mathbf{P} = \text{Stack}_{j=1}^{k}(\text{onehot}(i_j, n), \dim = 0) = \begin{bmatrix} \text{onehot}(i_0, n) \\ \text{onehot}(i_1, n) \\ \vdots \\ \text{onehot}(i_{k-1}, n) \end{bmatrix}. \tag{3}$$

In the above equation, $(i_0, i_1, \ldots, i_{k-1})$ can be thought as the indices of the token matrix. In line with digital circuit (Harris & Harris, 2012), machine learning, and its programming language's conventions (Paszke et al., 2019), we define onehot is a function that outputs a vector (one-dimensional tensor) containing binary values in $\{0,1\}$. Our definition of onehot function takes two arguments: index $i$ and length of the output vector $n$. The $i^{th}$ value of this output vector is one and the rest are zeros. For clarity, we give an example (index counts from 1): onehot$(2,3) = (0,1,0)$. There are two types of simple sampling mechanisms: sampling with replacement and sampling without replacement. For an operator to be a sampling without replacement operator, the indices must be unique, i.e. $i_j \neq i_k \ \forall \ j \neq k$.

We sample the vectors based on importance scores: the vectors' importance scores represent the likeliness of being picked. This implies that our method does not need to specify the number of sampled tokens, which is more versatile than learned projection matrices explored by previous method (Wang et al., 2020). The number of parameters of our importance score-based sampling is also smaller than projection matrices: $d \times 1$ versus $k \times n$, where $d, n, k$ are the number of token dimensions, tokens, and sampled tokens (or projected dimensions), respectively. However, setting a constant number of sampled tokens allows better parallelization, supported by PyTorch's compile method that makes code into a singular optimized kernel. The importance scores are computed via a learnable linear transformation: Linear : $\mathbb{R}^m \to \mathbb{R}$. We use Gumbel distribution to include randomness to the importance scores, similar to (Jang et al., 2017). The Gumbel distribution probability density function (PDF) and importance score computation is given as follows:

$$\varphi(x) = e^{-(x+e^{-x})},$$
$$\boldsymbol{z} = \text{Linear}_{\mathbf{S}}(\mathbf{X}) = \mathbf{X}W_S + B_S.$$

where $\varphi : \mathbb{R} \to \mathbb{R}$ is the PDF of the Gumbel distribution.

To estimate gradients from non-differentiable choosing operator, Gumbel-Softmax (Jang et al., 2017) has used reparameterization trick. In the forward pass, it returns hard-choose results using argmax, and in the backward pass, it uses a differentiable proxy for the gradient, as described in Equations 5 and 6, respectively. In Equation 6, $\tau$ controls the trade-off between the gradient variance and the optimizability of hardmax function. As $\tau$ transits from $0^+$ to $+\infty$, samples from Equation 6 transits from a categorical distribution (accurate gradients but vanishing) to a uniform distribution (strong gradient signal but inaccurate gradients). While this causes discrepancies between the forward and backward pass Jang et al. (2017), it provides a cheap and reliable enough method to optimize for categorical stochastic variables. The procedure to choose one vector from value matrix is given in the following equations:

$$\mathbf{V} = \mathbf{X}W_V + B_V \tag{4}$$
$$s(\mathbf{V}, \boldsymbol{z}, \boldsymbol{g}) = \mathbf{V}[\text{argmax}_i(\boldsymbol{z}[i] + \boldsymbol{g}[i])]. \tag{5}$$
$$\nabla s(\mathbf{V}, \boldsymbol{z}, \boldsymbol{g}) = \nabla \left( \sum_{i=1}^{n} \frac{\exp\left(\frac{\boldsymbol{z}[i] + \boldsymbol{g}[i]}{\tau}\right)}{\sum_{j=1}^{n} \exp\left(\frac{\boldsymbol{z}[j] + \boldsymbol{g}[j]}{\tau}\right)} \mathbf{V}[i] \right), \tag{6}$$

where $s, \mathbf{V}, \boldsymbol{z}, \boldsymbol{g}$ and $\tau$ are the sampling function, the value matrix, the score vectors, Gumbel sample vector, and the temperature scaling respectively; $\nabla$ denotes the gradient operator.

The problem is that Gumbel-Softmax gives off very small gradients due to its exponential nature. This is fine for its intended usage: to be put at the last layer or in a few specialized modules, but not a vital component in repeating sequential model cells. We can linearize (or polynomialize to be precise) the probability formula by raising the Softplus-ed score to the power of $\tau^{-1}$, $\tau \in \mathbb{R}^+$ instead of dividing the score by $\tau$ to control the trade-off. The following equations give the formulation of reparameterization trick via Softplus:

$$\text{Softplus}(x) = \log(1 + \exp(x)), \tag{7}$$
$$s(\mathbf{V}, \boldsymbol{z}, \boldsymbol{g}) = \mathbf{V}[\text{argmax}_i(\boldsymbol{z}[i] + \boldsymbol{g}[i])]. \tag{8}$$
$$\nabla s(\mathbf{V}, \boldsymbol{z}, \boldsymbol{g}) = \nabla \left( \sum_{i=1}^{n} \frac{\text{Softplus}\left(\boldsymbol{z}[i] + \boldsymbol{g}[i]\right)^{\tau^{-1}}}{\sum_{j=1}^{n} \text{Softplus}\left(\boldsymbol{z}[j] + \boldsymbol{g}[j]\right)^{\tau^{-1}}} \mathbf{V}[i] \right). \tag{9}$$

From here, we can derive the trivial sampling with replacement procedure by repeatedly applying Equation 6 or Equation 8, as follows:

$$\boldsymbol{z} = \text{Linear}_{\mathbf{S}}(\mathbf{X}) = \mathbf{X}W_S + B_S. \tag{10}$$

$$\text{Sam}(\mathbf{X}, k) = \text{Stack}_{i=1}^{k}\left(s(\mathbf{X}, \boldsymbol{z}, \boldsymbol{g}_i), \dim = 0\right). \tag{11}$$

where each vector $\boldsymbol{g}_i$ contains $n$ independent samples from the Gumbel distribution. $s$ is a stochastic function returning a (different) token each time it is called (in forward pass), defined in Equation 5 or Equation 8. Stacking the sampled vectors along the first dimension creates the sampled token matrix. While we do not use sampling with replacement in our method, for training stability, we found that this method requires gradients passing through the sampling function to be multiplied by $\frac{1}{k}$.

The drawback of sampling with replacement is repeating sampled vectors. Essentially, this causes the real sampling rate to be low despite even at a high number of samples, greatly limits model capacity. Recognizing this limitation of sampling with replacement, we construct a data-driven differentiable sampling without replacement that is parallelizable, in contrast to existing sequential ones. Our method sorts the vectors by importance scores, selects $k$ vectors with the highest scores in set $\mathcal{S}_1$ and $k$ random vectors in set $\mathcal{S}_2$, satisfying this constraint: $\mathcal{S}_1 \cap \mathcal{S}_2 = \emptyset$, concatenates set $\mathcal{S}_1$ and set $\mathcal{S}_2$ into set of $k$ binary bins $\mathcal{S}$, and finally selecting $k$ vectors using Equation 8 and Equation 9. While the sorting procedure is not differentiable, the whole procedure ensures the important tokens receive higher scores than the unimportant ones iteratively through a Softplus probability bin and gradient descent process. In practice, we find the resulting sampled tokens to be duplets (linear combinations of a token from $\mathcal{S}_1$ and a token from $\mathcal{S}_2$) is good enough and much easier to optimize. Therefore, we accept the noise from the components from $\mathcal{S}_2$ instead of true sampling with reparameterization trick, which results in a lower convergence rate and requires intensive hyperparameter tuning of the $\tau$ parameter. $\tau$ is set as 1.0 throughout all experiments. The algorithm is described in Figure 2.

With relative positional encodings, sparse and dense, the method to sample is similar. To inject relative positional encoding densely (e.g. injecting Euclidean Distance Matrix information onto attention maps; injecting token positional encodings), it is as simple as concatenating the embeddings reserved for relative positional encoding (e.g. point cloud coordinates) with the input of sampler function. This is possible if assuming there exists a function that linearly or non-linearly decomposes the relative positional encoding matrix with shape $(n, n)$ into two matrices with shape $(n, d)$. The sampler relies solely on the contextual meaning of token embeddings and does not depend on the relative positional information. This is by design and based on an assumption (which should be observed in Figure 3 and Figure 8): relative positional information gradually transmits to token embeddings through transformer layers. Sparse relative positional encoding; however, is much more complicated to be implemented for parallel processing. We provide a linear computational complexity algorithm that can be run on CPU (or PyTorch Sparse Tensor, but not efficient enough to justify):

- Step 1: Sample from the token embeddings using pseudocode provided in Figure 2.

- Step 2: Get the indices and probability score (the weight of binary linear combinations) of the top-k tokens and the randomly sampled tokens.

- Step 3: Construct two relative positional matrices $\mathbf{A}_{\textbf{top}}, \mathbf{A}_{\textbf{rand}}$ with shape $(n, k)$ where $n, k$ are number of tokens and number of sampled tokens, respectively.

- Step 4: Fill $\mathbf{A}_{\textbf{top}}, \mathbf{A}_{\textbf{rand}}$ with $\mathbf{X}_{\mathbf{R}}[\text{indices}]$. $\mathbf{X}_{\mathbf{R}}$ here is a sparse matrix data structure.

- Step 5: Transform each relative positional matrix through needed learnable layers provided in Section 4.3. It is done this way to avoid the layer transforming the relative features ended up learning the noisy relative features.

- Step 6: Multiplied the two matrices $\mathbf{A}_{\textbf{top}}, \mathbf{A}_{\textbf{rand}}$ with their corresponding probability score provided by Step 2.

- Step 7: Return $\mathbf{A}_{\textbf{top}} + \mathbf{A}_{\textbf{rand}}$

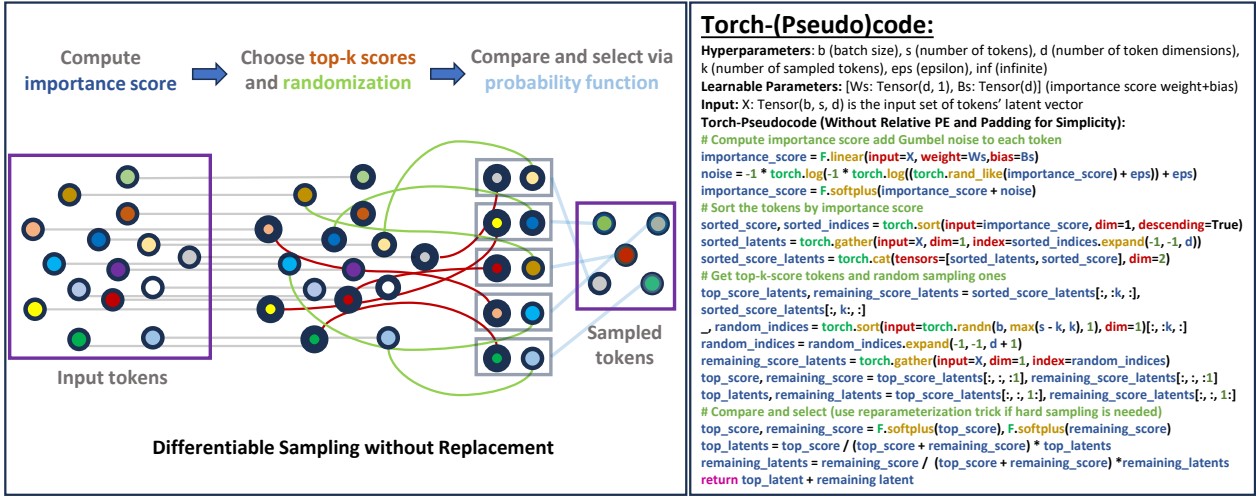

Figure 2: Our proposed differentiable sampling without replacement.

In practice, we treat the sparse relative positional matrix $\mathbf{X_R}$ as a dense matrix/tensor for parallelizability (and to be compiled to a singular kernel via compile from PyTorch method). This is a limitation of our work in engineering.

## 4.3 Attention Matrix Construction

To recall, any generic self-attention is expressed as follows:

$$
\begin{aligned}
\mathbf{Q} &= \text{Linear}_{\mathbf{Q}}(\mathbf{X}) = \mathbf{X}W_Q + B_Q, \\
\mathbf{K} &= \text{Linear}_{\mathbf{K}}(\mathbf{X}) = \mathbf{X}W_K + B_K, \\
\mathbf{V} &= \text{Linear}_{\mathbf{V}}(\mathbf{X}) = \mathbf{X}W_V + B_V, \\
\text{Attn}(\mathbf{X}) &= \mathbf{X} + \text{Prob}\left(\text{Score}(\mathbf{Q}, \mathbf{K})\right)\mathbf{V}.
\end{aligned}
$$

In vanilla transformer (Vaswani et al., 2017), function Prob (probability function / attention matrix scale normalizer based on number of tokens) and Score (attention nonlinearity) are softmax and scaled-dot product. We provide formulate as one attention head with reasons given in Section 3. $\mathbf{Q}, \mathbf{K}, \mathbf{V}$ are matrices of size $n \times d$. $n, d$ in the following equations are the number of tokens and the number of token dimensions, respectively. The equations are expressed as follows:

$$
\text{Prob}(\mathbf{A})[i, j] = \frac{\exp(\mathbf{A}[i, j])}{\sum_{k=1}^{n} \exp(\mathbf{A}[i, k]) + \epsilon}, \tag{12}
$$

$$
\text{Score}(\mathbf{Q}, \mathbf{K})[i, j] = \frac{\sum_{k=1}^{d}(\mathbf{Q}[i, k] * \mathbf{K}[j, k])}{\sqrt{d}}, \tag{13}
$$

$$
\text{Attn}(\mathbf{X}) = \mathbf{X} + \text{Prob}(\text{Score}(\mathbf{Q}, \mathbf{K}))\mathbf{V}, \tag{14}
$$

with Equation 12 and Equation 13 constructs Equation 14 using Equation 2. Subsequent attention construction may only specify the function Prob and the function Score.

It is well-known that transformer is hard to optimize (Liu et al., 2020). We propose a conjecture that the ease of network training is strongly related to the componentwise convexity and the backpropagated gradient scale. This is based on a stark difference between transformers and network architectures: convolutional neural networks and multi-layer perceptrons are all pseudoconvex but self-attention modules are not. Here,

we reformulate the attention matrix such that for a single layer, the output is pseudoconvex with respect to the weights.

We discovered the pairwise Maxout attention nonlinearity (derived from the Maxout activation (Goodfellow et al., 2013)) is convex. When combined with the attention module (including the feed-forward network), the whole transformer layer is pseudoconvex, with informal proof in Section 4.4 and formal one in Appendix B.2. The following equations express leaky factor, ReLU-probability function, and the pairwise Maxout attention nonlinearity (takes element-wise maximum, sum over the values, and scale by $\sqrt{d}$), respectively:

$$\mathbf{C} = \text{Linear}_{\mathbf{C}}(\mathbf{X}) = \mathbf{X}W_C + B_C,$$

$$\text{Score}(\mathbf{Q}, \mathbf{K})[i, j] = \frac{\sum_{k=1}^{d} \max(\mathbf{Q}[i, k], \mathbf{K}[j, k])}{\sqrt{d}},$$

$$\text{Prob}(\mathbf{A}, \mathbf{C})[i, j] = \frac{\text{ReLU}(\mathbf{A}[i, j])}{\sum_{r=1}^{n} \text{ReLU}(\mathbf{A}[i, r]) + \text{Softplus}(\mathbf{C}[r, 1]) + \epsilon}$$

$$\text{Attn}(\mathbf{X}) = \mathbf{X} + \text{Prob}(\text{Score}(\mathbf{Q}, \mathbf{K}), \mathbf{C})\mathbf{V}.$$

While the leaky factor is initially introduced for numerical stability, it allows better relative information aggregation. In practice, we use a linear combination for relative positional encoding alongside the leaky probability function. The relative positional encoding matrix receives different linear transformations projecting $n$-dimensional-vectors into $h$-dimensional-vectors for each layer. With $\mathbf{X_R}$ being the relative positional encoding matrix, the following equations express how we incorporate relative positional encoding into attention scores:

$$\mathbf{R}_{\text{mul}} = \text{Softplus}(\text{Linear}_{\mathbf{R}_{\text{mul}}}(\mathbf{X_R})),$$

$$\mathbf{R}_{\text{add}} = \text{Linear}_{\mathbf{R}_{\text{add}}}(\mathbf{X_R}),$$

$$\text{Score}(\mathbf{Q}, \mathbf{K})[i, j] = \frac{\sum_{k=1}^{d} \max(\mathbf{Q}[i, k], \mathbf{K}[j, k])}{\sqrt{d}} * \mathbf{R}_{\text{mul}}[i, j] + \mathbf{R}_{\text{add}}[i, j],$$

$$\text{Prob}(\mathbf{A}, \mathbf{C})[i, j] = \frac{\text{ReLU}(\mathbf{A}[i, j])}{\sum_{r=1}^{n} \text{ReLU}(\mathbf{A}[i, r]) + \text{Softplus}(\mathbf{C}[r, 1]) + \epsilon}$$

$$\text{Attn}(\mathbf{X}) = \mathbf{X} + \text{Prob}(\text{Score}(\mathbf{Q}, \mathbf{K}), \mathbf{C})\mathbf{V}.$$

**Leakiness injects rank.** For any non-leaky probability function, the self-attention fails to distinguish any two tokens of a rank-1 uniform input, commonly known as the over-smoothing phenomenon (Noci et al., 2022; Dong et al., 2023). This case proves relative-positional-encoding-based (RPE-based) transformers' approximation power is far from being universal (Luo et al., 2022) as opposed to absolute-positional-encoding-based (APE-based) Transformer (Yun et al., 2020). This barrier limits transformers' performances in relative positional data structures such as graphs and point clouds. In contrast, with the addition of the positive leaky component $\mathbf{C}$, we empirically show that the rank progression of token representation through our RPE-based Transformer gradually gains some rank. We will name this phenomenon **rank injection**. To visualize rank injection, in Figure 3, we use 30 samples of random weights to measure the rank progression of token representation with one random normal relative matrix over the layers.

### 4.4 Theoretical Analysis Summary

In order to theoretically justify the effectiveness of our proposed model, we will analyze its convexity power. Even though rigorously, it is not possible for probability distribution functions to exhibits actual convexity (Tsurumi, 1966), a weaker yet significant function class that also draw heavy attention in optimization theory (Liu et al., 2012) (Qin et al., 2016) are **pseudoconvex functions**, defined as follows:

**Definition 4.1.** Let $\nabla$ denote the gradient operator and $\mathcal{S} \subset \mathbb{R}^n$ is a convex open set, a function $f : \mathcal{S} \to \mathbb{R}$ is said to be pseudoconvex in $\mathcal{S}$ if for any $x_1, x_2 \in \mathcal{S}$ such that $\langle \nabla f(x_1), x_2 - x_1 \rangle \geq 0$, then $f(x_2) \geq f(x_1)$.

Pseudoconvex functions hold a very important lemma, which states that **every stationary point is also a global minimizer**. To visualize this concept, we sketched two functions with convex and pseudoconvex

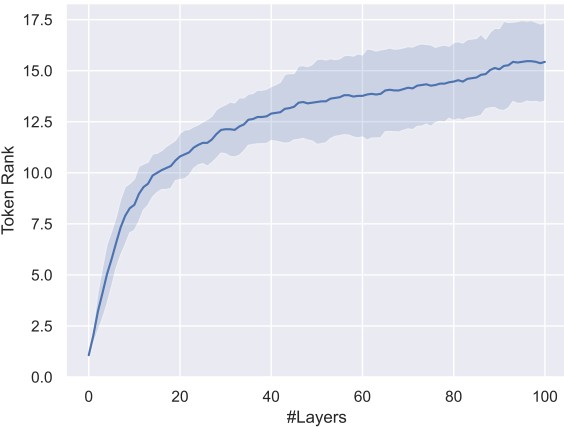

Figure 3: Rank progression of token representation with 256 tokens and embedding size 512 through 100 randomly initialized single head SFT layers with leaky probability function and pairwise maxout attention nonlinearity.

properties respectively in Figure 4. In application to deep learning, pseudoconvexity not only tackle tricky challenges in optimization such as saddle points and local minimas but also improve the overall performance of artificial neural networks.

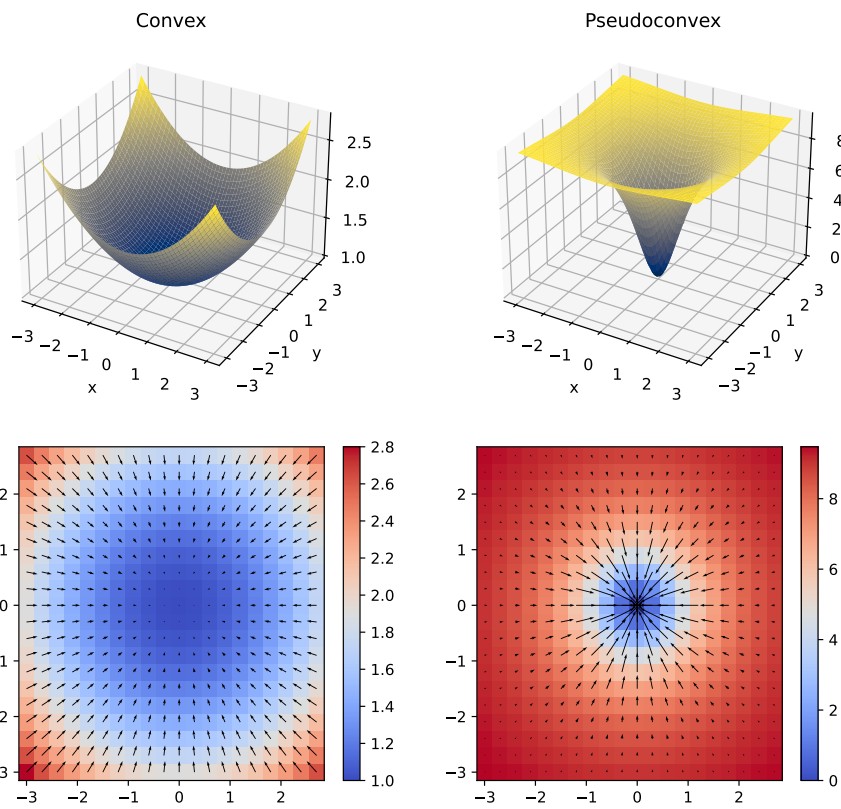

Figure 4: Comparison between a convex function $z = \frac{x^2+y^2+10}{10}$ (top-left) and a pseudoconvex function $z = \frac{10(x^2+y^2)}{x^2+y^2+1}$ (top-right) and their corresponding heatmaps with gradient vector fields (bottoms). The pseudoconvex function greatly resemble its convex counterpart regarding the search for the local minima.

In summary of our theoretical analysis, we will first show the convex inefficiency of two well-known attention settings, namely the vanilla attention (Vaswani et al., 2017) and ReLU-based attention with dot product (Shen et al., 2023) by handpicking some representative counterexamples. The details of the statements and corresponding proofs are provided at Appendix Section B.1.

Next, we will rigorously show that **SFT is pseudoconvex with both linear and GeLU activation in FFN layer**. Informally, we present the result as follows.

**Theorem 4.2.** *(informal) The SFT layer with linear or GeLU FFN activation and no sampling is componentwise pseudoconvex with respect to certain combinations of weights.*

We provide the full theorem and proof in Appendix Section B.2.

In compensation to the non-decreasing gradient behaviour of pseudoconvex functions comparing to convex counterparts, we also derived an adaptive lower bound and global upper bound of the expectation of the Frobenius norm of the gradients of SFT, which is often referred as gradient norms for short, and analyze their complexity with respect to the number of tokens. The boundedness of the gradient norms should somewhat represent their magnitude and address the robustness of SFT against challenges such as sharp/flat points, i.e. places at which the gradients are too small/large.

**Theorem 4.3.** *(informal) Let $\boldsymbol{Sa} = \boldsymbol{Sa}(W_Q, W_K, W_V)$ be the SFT attention layer output in Algorithm **??**, then $\mathbb{E} \left\| \frac{\partial \boldsymbol{Sa}}{\partial W_V} \right\|_F^2$ and $\mathbb{E} \left\| \frac{\partial \boldsymbol{Sa}}{\partial W_Q} \right\|_F^2$ has complexity $\Theta(n)$ and $\Theta(1)$ respectively, where $n$ is the number of tokens.*

Since the full proof for Theorem 4.3 is quite convoluted, we would like to sketch the proof as follows:

- Adaptive lower bound:
    - **Step 1:** Derive a fractional function form $f(\boldsymbol{W})/g(\boldsymbol{W})$ where $\boldsymbol{W}$ represents the set of parameters,
    - **Step 2:** Convexify the fractional form via the convex envelope derived by (Tawarmalani & Sahinidis, 2002),
    - **Step 3:** Evaluate the lower bound using the Jensen inequality,
    - **Step 4:** Make any simplification if necessary.

- Global upper bound: employ the norm product inquality and the Jensen inequality for concave functions.

The full statement and proof is at the Appendix Section B.3.

### 4.5 Complexity Analysis

Consider the input token $\mathbf{X} \in \mathbb{R}^{n \times d}$, query and key weights $W_Q, W_K \in \mathbb{R}^{d \times d_a}$.

The computational cost for each step is as follows:

- Linear transformations for $\mathbf{Q}, \mathbf{K}$ and $\mathbf{V}$: $O(nd^2)$,

- Importance score of the queries and keys: $O(nd)$,

- Selecting top-$k$ importance score: $O(n \log n)$ with sorting method or $O(n)$ with quickselect algorithm,

- Attention between $n$ queries and top-$k$ most important keys: $O(nkd_a)$,

- FFN network: $O(nd^2)$.

Given these complexities and considering the long-ranged data structures which can implies $n \gg d$, the asymptotic computational complexity of one single SFT layer is $O(n)$. The asymptotic space complexity of one single SFT layer is also $O(n)$.

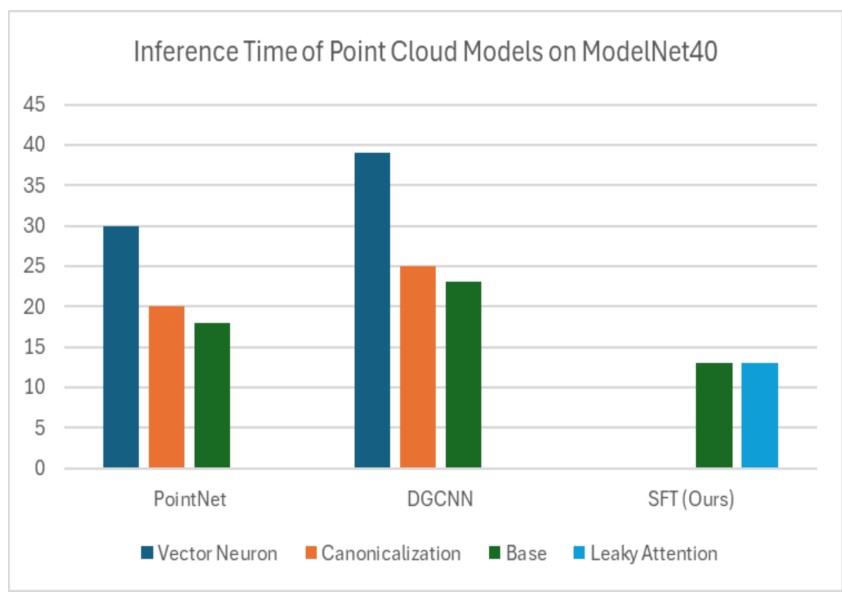

Figure 5: Inference time (in seconds) of the networks for ModelNet40 classification test split in 1 A100 GPU and 8 CPUs with a batch size of 32. The Vector Neuron (Deng et al., 2021) and Canonicalization (Kaba et al., 2023) framework is applied in PointNet (Charles et al., 2017) and DGCNN (Wang et al., 2019). The leaky attention function is applied to our SFT model. Our model in this plot does not use compile method from PyTorch to speed up for fair comparison (when using, it halves the inference time on A100). The results of others are taken from (Kaba et al., 2023).

## 5 Experiments

### 5.1 Objectives

The core of any foundational model is its efficacy in modeling diverse data modalities. And, the fundamental difference between data modalities is the different types of relationships between tokens. Therefore, we conduct extensive experiments to measure the ability of our attention mechanism to model dense low-rank (point clouds) and sparse high-rank relationships (graphs and sequences). As we propose our pseudoconvex mechanism, we also conduct a comparative experiment to measure the real runtime of our transformer formulation against the vanilla transformer.

In short, we answer three following questions:

- How effectively can leakiness encode relative information into the tokens' representation?

- How effectively can our sampling-based global attention model long-range interaction?

- How significant does our pseudoconvex formulation aid model convergence rate?

### 5.2 Relative Learning

We conduct experiments of relative learning on two particular archetypes: Dense Low-Rank and Sparse High-Rank because most data that exists in reality falls in either of the two mentioned groups.

#### 5.2.1 Dense Low-Rank relative Learning

To measure the ability of our model in modeling dense low-rank relationship, we test our model on two point cloud tasks involving two datasets: Object Classification on ModelNet40 (Wu et al., 2015), Part Segmentation

on ShapeNetPart (Chang et al., 2015); with/without rotational invariant constraint. The motivation we choose point cloud to experiment Dense Low-Rank relative Learning is simple: rank of Euclidean Distance Matrix of $p$-dimensional points is at most $p+2$ (Gower, 1985) and point cloud coordinates can be reconstructed from their Euclidean Distance Matrix through Multidimensional Scaling algorithm.

Our experiments on rotational invariant constraint specifically remove all of point cloud coordinates in input tokens, which forces our model to use only relative information to discriminate. By making our model rely solely on relative information, the experiment validates the generalizability of information obtained through rank injection phenomena. We further compare it against other dedicated rotational invariant models in Table 1 and it shows comparable performance to these dedicated rotational invariant methods, while can be inferenced **hundreds of percents** faster (computational time measured in Figure 5). Note that, our relative positional encoding scheme **can be dropped into any point cloud transformer**, making it rotational invariant with neglectable additional cost (because all of the point cloud transformers already feature relative positional encoding).

We also conduct experiments on point cloud datasets without rotational invariant constraints to measure how effective our method as a whole is in modeling non-sequential data in Table 2. It shows competitive results against other point cloud models. This is because our model does not include any data-modality-specific inductive bias (like the usage of kNN for locality heuristic) and is susceptible to overfitting in scenarios where data is not abundant.

Table 1: Experimental results to measure the effectiveness of leaky probability function in terms of transferring relative information to tokens' representation. The table shows shape classification results on ModelNet40 dataset (Wu et al., 2015) and part segmentation results on ShapeNetPart dataset (Chang et al., 2015) under rotational invariant constraint with popular baselines in rotational invariance/equivariant literature. z/z, z/SO(3), SO(3)/SO(3) signifies the model trained with 3D coordinates (no rotational invariant constraint), the model trained with rotational invariant features/trained using rotational equivariant transformation, and trained with random rotation data augmentation, respectively.

| Method | ModelNet40 | | ShapeNetPart | |
|---|---|---|---|---|
| | $z/z^{\uparrow}$ | $z/SO(3)^{\uparrow}$ | $z/SO(3)^{\uparrow}$ | $SO(3)/SO(3)^{\uparrow}$ |
| SFCNN (Rao et al., 2019) | 91.4 | 84.8 | - | - |
| TFN (Thomas et al., 2018) | 88.5 | 85.3 | 76.8 | 76.2 |
| RI-Conv (Zhang et al., 2019) | 86.5 | 86.4 | 75.3 | 75.3 |
| SPHNet (Poulenard et al., 2019) | 87.7 | 86.6 | - | - |
| ClusterNet (Chen et al., 2019) | 87.1 | 87.1 | - | - |
| GC-Conv (Zhang et al., 2020) | 89.0 | 89.1 | 77.2 | 77.3 |
| RI-Framework (Li et al., 2022) | 89.4 | 89.4 | 79.2 | 79.4 |
| VN-PointNet (Deng et al., 2021) | 77.5 | 77.5 | 72.4 | 72.8 |
| VN-DGCNN (Deng et al., 2021) | 89.5 | 89.5 | 81.4 | 81.4 |
| CN(NL)-PointNet (Kaba et al., 2023) | 79.9 | 79.6 | 73.5 | 73.6 |
| CN(NL)-DGCNN (Kaba et al., 2023) | 88.7 | 88.8 | 78.4 | 78.5 |
| SFT (Ours) | 91.1 | 87.2 | 78.3 | - |

### 5.2.2 Sparse High-Rank relative Learning

To measure the ability of our model in modeling sparse high-rank relationship, we test our model on two data modalities: sequence and graph. For sequential tasks, we choose Long-Range-Arena Benchmark (Tay et al., 2021), a standard to measure the effectiveness of efficient transformer schemes in sequential modeling. For graph tasks, we choose three datasets: Peptides-func (Singh et al., 2015), Peptides-struct (Singh et al., 2015), and PascalVOC-sp (Everingham et al., 2010) in Long-Range-Graph-Benchmark (Dwivedi et al., 2022b), which is the equivalent of Long-Range-Arena-Benchmark for graph transformers.

The reason we chose these two data modalities to represent sparse high-rank relative learning is intuitive: both the causal relationship of sequences and graph adjacency matrix is very sparse yet high rank. The competitive

Table 2: Experimental results to measure the effectiveness of our sparse attention on non-sequential data structure, particularly spatial data structure. The table shows shape classification results on ModelNet40 dataset (Wu et al., 2015) and part segmentation results on ShapeNetPart dataset (Chang et al., 2015) along with popular baselines in the point cloud analysis literature.

| Method | ModelNet40 | | ShapeNetPart | |
|---|---|---|---|---|
| | mAcc$^\uparrow$ | OA$^\uparrow$ | c. IoU$^\uparrow$ | i. IoU$^\uparrow$ |
| PointNet (Charles et al., 2017) | 86.2 | 89.2 | 80.4 | 83.7 |
| Set Transformer (Lee et al., 2019) | - | 90.4 | - | - |
| PointNet++ (Qi et al., 2017) | - | 91.9 | 81.9 | 85.1 |
| SpecGCN (Wang et al., 2018) | - | 92.1 | - | - |
| PointCNN (Li et al., 2018) | 88.1 | 92.2 | 84.6 | 86.1 |
| DGCNN (Wang et al., 2019) | 90.2 | 92.2 | 82.3 | 85.1 |
| PointWeb (Zhao et al., 2019) | 89.4 | 92.3 | - | - |
| SpiderCNN (Xu et al., 2018) | - | 92.4 | 81.7 | 85.3 |
| PointConv (Wu et al., 2019) | - | 92.5 | 82.8 | 85.7 |
| Point2Sequence (Liu et al., 2019) | 90.4 | 92.6 | - | 85.2 |
| KPConv (Thomas et al., 2019) | - | 92.9 | 85.1 | 86.4 |
| InterpCNN (Mao et al., 2019) | - | 93.0 | 84.0 | 86.3 |
| Point Transformer (Zhao et al., 2021) | 90.6 | 93.7 | 83.7 | 86.6 |
| Sequoia (Trang et al., 2024) | 88.4 | 92.0 | 80.6 | 83.8 |
| SFT (Ours) | 88.2 | 91.1 | 81.3 | 84.5 |

experimental results of our model in Long-Range-Arena-Benchmark and Long-Range-Graph-Benchmark are shown in Table 4 and Table 3, respectively.

In sequential tasks, our model has competitive results against many other efficient transformers and the full attention transformer. However, it should be note that we did not reach the performance level of models designed specifically for sequential tasks like MEGA (Ma et al., 2023) and S4 (Gu et al., 2022). The inductive bias of EMA (Exponential Moving Average) in MEGA and SSM cannot be applied to other data structures like point clouds and graphs.

In graphs tasks, our model has relatively good results in peptides dataset compared to other efficient transformers, full attention transformer, and superior results to all classical message passing neural networks. However, since our transformer construction for graph is simple and does not rely on local information aggregation like GraphGPS (Rampášek et al., 2022), it does not perform well on PascalVOC-sp dataset. This shows a limitation on how our method's relative information aggregation.

### 5.2.3 Performance Gap between Relative and Absolute Positional Information Aggregation

Additionally, we compare two variants of our models (with/without absolute position) on both Dense-Low-Rank and Sparse-High-Rank experiments. We notice a significant performance gap on all of the experimented datasets: ModelNet40, ShapeNetPart, Peptides-func, Peptides-struct, PascalVOC-sp; shown in Table 5. It suggested that our relative modeling performance is inferior to introducing relations directly into token embeddings, despite our formulation allowing more complicated constraints to be implemented.

We suspect this is related to how we construct relative information since we only use two linear layers to transform relative information. We have experimented with multi-layer perceptron to transform relative information, but it proved to be too slow in practice. This is because transforming $k * n$ vectors (or an entire attention map), where $k$ and $n$ is the number of sampled tokens and the number of tokens, respectively, is an extremely costly operation. We believe efficient relative information transformation can be an interesting open question for future research.

Table 3: (Long Range Graph Benchmark) Performance on Peptide and Computer-Vision Graph datasets. The performance of our model on peptide-func, peptide-struct, and the computer vision-based PascalVOC datasets is measured using Average Precision (AP), Mean Absolute Error (MAE), and F1-score metrics, respectively.

| Method | Pept-func $\uparrow$ | Pept-struct $\downarrow$ | PasVOC-sp $\uparrow$ |
|---|---|---|---|
| GCN (Kipf & Welling, 2017) | 0.5930 | 0.3496 | 0.1268 |
| GCNII (Chen et al., 2020) | 0.5543 | 0.3471 | 0.1698 |
| GINE (Brossard et al., 2021) | 0.5498 | 0.3547 | 0.1265 |
| GatedGCN (Bresson & Laurent, 2018) | 0.5864 | 0.3420 | 0.1265 |
| GatedGCN (RWPE) (Bresson & Laurent, 2018) | 0.6069 | 0.3357 | 0.1265 |
| Transformer (LapPE) (Vaswani et al., 2017) | 0.6326 | 0.2529 | 0.2694 |
| Transformer (RWPE) (Vaswani et al., 2017) | 0.6502 | 0.2620 | 0.2718 |
| SAN (LapPE) (Kreuzer et al., 2021) | 0.6384 | 0.2683 | 0.3230 |
| SAN (RWPE) (Kreuzer et al., 2021) | 0.6562 | 0.2545 | 0.3216 |
| GPS (Rampášek et al., 2023) | 0.6535 | 0.2500 | 0.3748 |
| Exphormer (Shirzad et al., 2023) | 0.6527 | 0.2481 | 0.3975 |
| GPS-Sequoia-RWPE (Trang et al., 2024) | 0.6755 | 0.2453 | 0.3379 |
| SFT (Ours) | 0.6674 | 0.2661 | 0.1961 |
| SFT-RWPE (Ours) | 0.6902 | 0.2655 | 0.2181 |

Table 4: (Long Range Arena) Accuracy on the full suite of long range arena (LRA) tasks. The performance of our model on peptide-func, peptide-struct, and the two computer vision-based datasets is measured using Average Precision (AP), Mean Absolute Error (MAE), and F1-score metrics, respectively.

| Method | ListOps | Text | Retrieval | Image | Pathfinder |
|---|---|---|---|---|---|
| BigBird (Zaheer et al., 2020) | 36.05 | 64.02 | 59.29 | 40.83 | 74.87 |
| Reformer (Kitaev et al., 2020) | 37.27 | 56.10 | 53.40 | 38.07 | 68.50 |
| Performer (Choromanski et al., 2021) | 18.01 | 65.40 | 53.82 | 42.77 | 77.05 |
| Linformer (Wang et al., 2020) | 35.70 | 53.94 | 52.27 | 38.56 | 76.34 |
| Luna-256 (Ma et al., 2021) | 37.98 | 65.78 | 79.56 | 47.86 | 78.55 |
| Transformer (Vaswani et al., 2017) | 36.37 | 64.27 | 57.46 | 42.44 | 71.40 |
| Sequoia (Trang et al., 2024) | 37.70 | 75.10 | 67.04 | 49.88 | 87.30 |
| S4 (Gu et al., 2022) | 88.65 | 76.02 | 87.09 | 86.09 | 86.05 |
| MEGA (Ma et al., 2023) | 63.14 | 90.43 | 91.25 | 90.44 | 96.01 |
| SFT-Relative (Ours) | 39.95 | 64.54 | 71.33 | 47.65 | 78.39 |

## 5.3 Learning Curve Analysis

We conduct learning curve analysis to see the effect of our pseudoconvex formulation to the convergence rate as well as the effect of sampling rate on the performance of our model. Since learning curve can be too noisy to interpret, we include curves of cumulative maximum attained performance for easier interpretation.

### 5.3.1 Convergence Speed

To verify our claim on the effectiveness of our componentwise pseudoconvex transformer module, we conduct comparative experiments between three models: the model using Leaky-Relu Probability function with Maxout Attention score (1), the model using Softmax Probability function with Scaled-Dot Attention score

Table 5: Performance Gap between Relative and Absolute Positional Information Aggregation

| | ModelNet40$^\uparrow$ | ShapeNetPart$^\uparrow$ | Peptides-func$^\uparrow$ | Peptides-struct$^\downarrow$ | PascalVOC-sp$^\uparrow$ |
|---|---|---|---|---|---|
| **relative** | 87.2 | 78.3 | 0.67 | 0.2661 | 0.20 |
| **Absolute** | 91.1 | 81.3 | 0.69 | 0.2655 | 0.22 |
| **Performance Gap** | 3.9 | 3.0 | 0.02 | -0.001 | 0.02 |

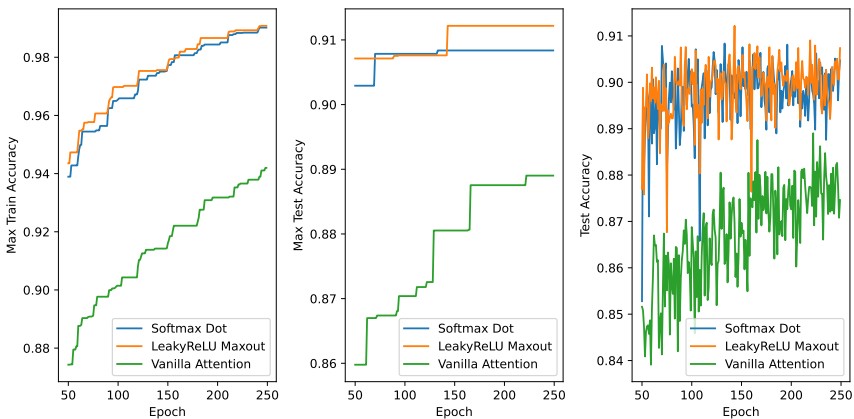

Figure 6: Comparison between learning curves of SFT with Leaky Relu-based + Maxout (orange), with softmax + dot product (blue) and vanilla transformer without relative information (green) on ModelNet40 throught 256 epochs. The actual test accuracy curve is shown (right) while the max accuracy through all epochs are also visualized during training (left) and testing (middle).

(2), and the vanilla transformer without relative information (3). The models are trained on ModelNet40 dataset. The comparison between (1) and (2) shows the effectiveness of the pseudoconvex formulation against the traditional softmax-dot-product. The comparison between (1, 2) and (3) shows the necessity to include relative information. The experiments are shown in Figure 6. The higher convergence rate of our model verifies the effectiveness of our pseudoconvex formulation against the vanilla attention formulation, and the better performance of the two models using relative information against vanilla attention verifies the usefulness of relative information.

### 5.3.2   Performance-Efficiency Tradeoff on Sampling Rate

To examine the performance-efficiency tradeoff, we trained our model with different sampling rates: 0.4%, 6.25%, 12.5%, 25%, and 50%, measured both of their train and test accuracy per epoch, and measured the runtime of models of different sampling rates.

In Figure 7, we show the result of SFT performance with five different sampling rates. Our results suggest that higher sampling rate is likely to increase the convergence speed and performance. However, it should be noted that the increment of sampling rate from 25% to 50% does not result in better test accuracy even has a clear higher sampling rate and an extremely low sampling rate (0.4%) can achieve reasonable performance.

To precisely measure the runtime of models, we run with one CPU thread. SFT layers with various numbers of sampled points are compared against the vanilla transformer (using the built-in implementation in PyTorch to measure). The input is a batch of 4 1024-length sequences. The result is shown in Table 6, which shows that our model is more efficient than the vanilla transformer when the sampling percentage is less than 50%. It should be note that this does not exclude the computation of MLP module within the transformer module and the built-in vanilla transformer does not support relative information. The computational cost of modules is discussed in Section 5.4.

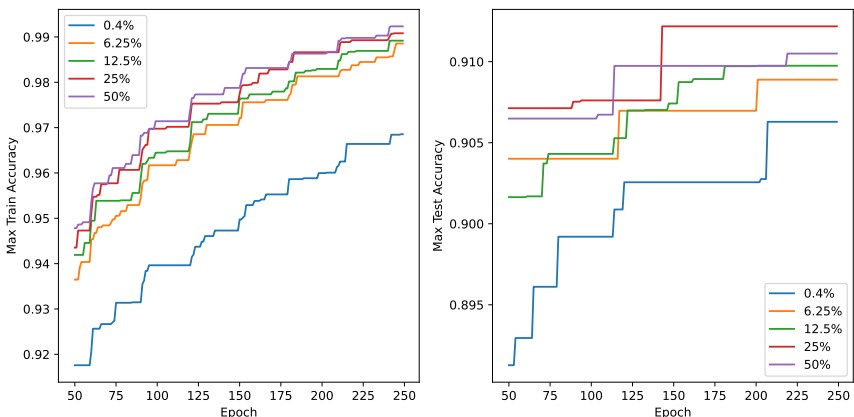

Figure 7: The influence of sampling rate onto the performance of SFT at five different sampling rates. The accuracy during training (left) and testing (right) are taken by max through 256 epochs.

Table 6: Model runtime measured in seconds (10 runs averaged).

| | SFT (Ours) | | | | | Transformer |
|---|---|---|---|---|---|---|
| **#Sampling** | 32 | 64 | 128 | 256 | 512 | 1024 |
| **%Sampling** | 3.13% | 6.25% | 12.5% | 25.0% | 50.0% | 100% |
| **Runtime↓** | 0.30s | 0.32s | 0.34s | 0.46s | 0.64s | 0.64s |

## 5.4 Computational Cost Breakdown

To provide a more comprehensive insight on the computational cost of submodules of an SFT layer, we measure the computational cost of each SFT submodule (measured in GFLOPS) with results in Table 7. The layer processes 1024 256-dimensional tokens.

From Table 7, it can be inferred that the sampling cost is negligible compared to other submodules. However, even with only two linear layers, the model is computational time for relative positional encoding module still has high computational cost. Without relative positional encoding, the model can be even faster; however, the application of our model is hindered (e.g. no longer being able to process graph data modality, performance drop on point cloud data modality).

Table 7: Submodule SFT Computational Cost

| Sampling rate | Sampling | QKVC | | | | MultiHeadAttention | | | MLP |
|---|---|---|---|---|---|---|---|---|---|
| | | Q | K | V | C | Relative | W_cat | Remainder | |
| 100% | 0.0037 | 0.268 | 0.268 | 0.268 | 0.016 | 0.268 | 0.268 | 0.044 | 2.2 |
| 50% | 0.0037 | 0.268 | 0.134 | 0.134 | 0.0084 | 0.134 | 0.268 | 0.022 | 2.2 |
| 25% | 0.0037 | 0.268 | 0.067 | 0.067 | 0.0042 | 0.067 | 0.268 | 0.009 | 2.2 |
| 12.5% | 0.0037 | 0.268 | 0.034 | 0.034 | 0.0021 | 0.034 | 0.268 | 0.004 | 2.2 |
| 6.25% | 0.0037 | 0.268 | 0.017 | 0.017 | 0.0011 | 0.017 | 0.268 | 0.002 | 2.2 |

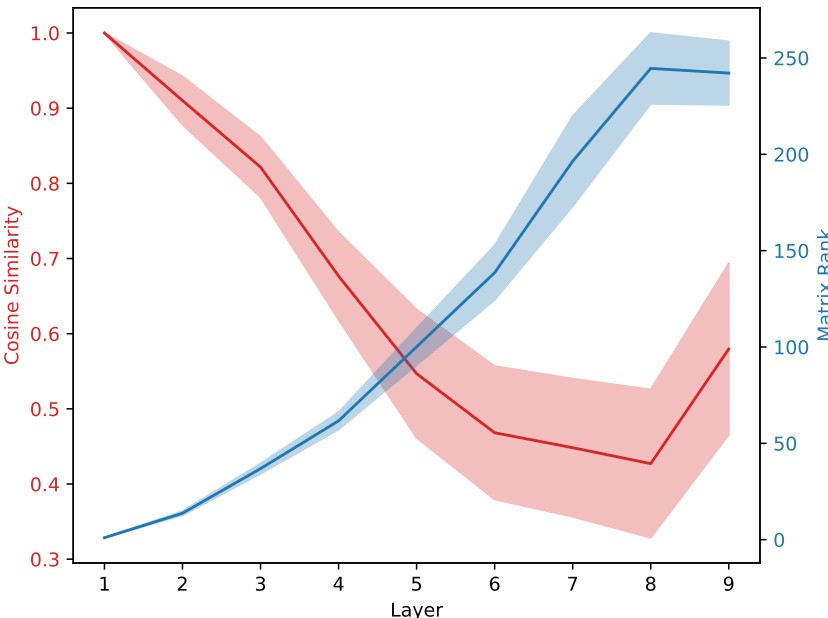

Figure 8: Similarity and Rank of Token Embeddings through layers of 32 Randomly Selected Data-Points

### 5.5 Rank Injection Phenomena

In Figure 3, we have explored how rank of input token embeddings gradually increases under random initialization. However, at a computationally feasible number of layers (1 - 20), the rank of input token embeddings is unfortunately still small and does not carry enough information to distinguish between different tokens. Here, we revisit the phenomena from a different perspective: a fully trained model. We measure cosine similarity between tokens and matrix rank layer by layer to show how information flows from relative positional encoding to token embeddings. The model and task we chose is rotational invariant point cloud classification on ModelNet40, which features 1024 256-dimensional tokens.

From Figure 8, it shows token embeddings start to be dissimilar around $5^{\text{th}}$ layer. Similarly, token embeddings' matrix rank rises through layer transformations, **near to the maximum possible rank** (the number of dimensions of token embeddings is 256). This shows the model can generate complex spatial relationships between tokens since the input relative positional encoding is low-rank (Gower, 1985).

## 6 Conclusion

In this paper, we made three contributions: a data-driven differentiable sampling without replacement method, self-attention formulation with pseudoconvexity and bounded gradient norm, and leaky probability function for relative positional encoding. Two of our three contributions aim at lowering the complexity of training transformers while the last one significantly improves the expressivity of transformer regarding relative positional encoding. We have empirically demonstrated the usefulness of the three modules and show at least one application: zero-additional computing cost to achieve rotational invariant - a desired property for multiple point cloud problems (e.g., molecular modeling). By easing the hyperparameter tuning pain, it may pave the way for very deep transformer networks to be trained at small facilities. In short, we make transformers more powerful and easier to train.

**Limitations and Future Works.** We list the limitations and the prospects of our work as the following:

- We have yet to incorporate dedicated point cloud modules into our model as well as use the leaky positional encoding for prominent state-of-the-art point cloud transformers.

- Our model's efficiency bottleneck lies on its relative positional encoding computation, where at full attention, the relative positional encoding takes as much as 50% of multi-head attention computational cost. Future work would be to incorporate more sophisticated relative positional encoding that works well on all data modalities (point clouds, sequences, images, graphs, ...).

- We have not yet experimented on heterogeneous datasets. Since SFT is a model that has shown effectiveness on a wide range of data modalities (point clouds, sequences, graphs), learning to process tokens of different types of data modalities is an interesting question. Processing tokens of different types of data modalities would provide unprecedented generative ability of models since it allows more precise information mediums such as graphs to describe relationships, tables to describe statistical data, and point clouds to describe 3D structures.

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

# A  Notations

**Numbers and Arrays**

$a$      A scalar

$\boldsymbol{a}$      A vector

$\mathbf{A}$      A matrix

$\mathbf{A}^{\top}$      Tranpose of matrix $\mathbf{A}$

$n$      The length of the input sequence

$d$      Embedding dimension

**Sets**

$\mathbb{R}^{a}$      Set of real vectors with length $a$

$\mathbb{R}^{a \times b}$      Set of real matrices with size $a \times b$

$[m]$      Set of all integers from 1 to $m$

$[a, b]$      Closed interval from $a$ to $b$

**Indexing**

$\boldsymbol{a}[i]$      $i^{th}$ element of vector $\boldsymbol{a}$, with indexing starts from 1

$\mathbf{A}[i, j]$      Element $(i, j)$ of matrix $\mathbf{A}$

$\mathbf{A}[:, j]$      Column $j$ of matrix $\mathbf{A}$

$\mathbf{A}[i]$      Row $i$ of matrix $\mathbf{A}$

$\mathbf{A}[: k]$      The first $k$ rows of matrix $\mathbf{A}$

$\mathbf{A}[k :]$      All rows from row $k$ of matrix $\mathbf{A}$

**Functions and Operators**

$\mathbf{A} \odot \mathbf{B}$      Element-wise Hamadart product between two matrices $\mathbf{A}$ and $\mathbf{B}$

$\mathbf{A}\mathbf{B}$      Matrix multiplication of two matrices $\mathbf{A}$ and $\mathbf{B}$

$\|x\|_{F}$      Frobenius norm of $x$

$\nabla$      Gradient operator

$\langle \boldsymbol{a}, \boldsymbol{b} \rangle$      Scalar product between vector $\boldsymbol{a}$ and $\boldsymbol{b}$

# B  Theoretical Analysis

## B.1  Nonconvexity of Other Attention Settings

As mentioned in Section 4.4, the definition of pseudoconvex functions is defined below.

**Definition B.1.** Let $\nabla$ denote the gradient operator and $\mathcal{S} \subset \mathbb{R}^{n}$ is a convex open set, a function $f : \mathcal{S} \to \mathbb{R}$ is considered pseudoconvex if for any $x_1, x_2 \in \mathcal{S}$ such that $\langle \nabla f(x_1), x_2 - x_1 \rangle \geq 0$ then $f(x_2) \geq f(x_1)$.

In this section, we will show the nonpseudoconvexity of two Transformer variations: the well-known vanilla Transformer (Vaswani et al., 2017) and our ReLU-based attention with dot-product (Shen et al., 2023). Specifically, we will point out some simplified counterexamples with well-defined inputs to achieve this goal.

Note that these counterexample can easily be generalized into arbitrary settings, but we handpicked them rather for simplicity.

**Theorem B.2.** *Let* $\mathbf{S}(W_Q, W_K) = \text{SM}(\mathbf{X}W_Q W_K^\top \mathbf{X}^\top)\mathbf{X}\overline{W_V} + \mathbf{X}$ *be the vanilla attention layer where* SM *is the softmax function with respect to query and key, then there exists a setting in which* $\mathbf{S}$ *is not pseudoconvex with respect to all pairs of weight entries.*

*Proof.* Let $\mathbf{X} \in \mathbb{R}^{3\times3}$, $W_Q, W_K \in \mathbb{R}^3$ and $\overline{W_V} \in \mathbb{R}^{3\times3}$. We consider a counterexample by letting $\mathbf{X} = \overline{W_V} = \mathbf{I}_3$. Calculating the output we have

$$\mathbf{S}(W_Q, W_K)[i,j] = \frac{e^{w_{Qi}w_{Kj}}}{\sum_{k=1}^3 e^{w_{Qi}w_{Kk}}} \qquad \forall i,j \in \{1,2,3\}.$$

where $w_{Qi}$ and $w_{Kj}$ is the $i^{th}$ and $j^{th}$ entry of the vectors $W_Q$ and $W_K$ respectively.

From here, we will prove that $\mathbf{S}$ is not quasiconvex, thus not pseudoconvex, with respect to all pairs of scalar weight entries.

- $\mathbf{S}$ is not quasiconvex with respect to the pair $(w_{Qs}, w_{Qr})$ for any $s \neq r$ and $s,r \in \{1,2,3\}$.

  Since we are considering convexity within the entries of the query, it is reasonable to fix $W_K = \overline{W_K} = \begin{pmatrix} 1 & 1 & 1 \end{pmatrix}^\top$ and consider the first entry $\mathbf{S}$, which is simply

  $$\mathbf{s}(w_{Q1}, w_{Q2}, w_{Q3}) = \frac{e^{w_{Q1}}}{e^{w_{Q1}} + e^{w_{Q2}} + e^{w_{Q3}}}$$

  Without loss of generality let us fix $w_{Q1} = \overline{w_{Q1}} = 0$ and consider the convexity with respect to the pair $(w_{Q2}, w_{Q3})$. By taking the sublevel set defined by some level $\alpha \in (0,1)$, one can easily observe that

  $$L_{\mathbf{s}}(\alpha) := \left\{ (w_{Q2}, w_{Q3}) \in \mathbb{R}^2 \quad \text{s.t} \quad e^{w_{Q2}} + e^{w_{Q3}} \geq \frac{1}{\alpha} - 1 \right\}$$

  is evidently not a convex set in two dimensional space. Therefore, $\mathbf{S}$ is not quasiconvex w.r.t $(w_{Q2}, w_{Q3})$.

- $\mathbf{S}$ is not quasiconvex with respect to the pair $(w_{Qs}, w_{Kr})$ for any $s,r \in \{1,2,3\}$.

  Without loss of generality, let $w_{Qs} = \overline{w_{Qs}} = w_{Ks} = \overline{w_{Ks}} = 0$ for $s \in \{2,3\}$ and consider the function representing the $[2,1]$ location entry of the $\mathbf{S}(W_Q, W_K)$ output

  $$\mathbf{s}(w_{Q1}, w_{K1}) = \frac{1}{e^{w_{Q1}w_{K1}} + 2}.$$

  Similarly, by taking the sublevel set defined by some level $\alpha \in (0,1)$, it is obvious that this set is not a convex set. Hence, the proof is complete.

$\square$

Not only vanilla attention fails to exhitbit the pseudoconvex property, but also attention settings with the presence of the dot product.

**Theorem B.3.** *Let* $\mathbf{R}(W_Q, W_K) = \text{PR}(\mathbf{X}W_Q W_K^\top \mathbf{X}^\top)\mathbf{X}\overline{W_V} + \mathbf{X}$ *be the ReLU based attention with dot product where*

$$\text{PR}(\mathbf{A})[i,j] = \frac{\text{ReLU}(\mathbf{A}[i,j])}{\sum_{k=1}^n \text{ReLU}(\mathbf{A}[i,k])},$$

*then there exists a setting in which* $\mathbf{R}(W_Q, W_K)$ *is not pseudoconvex w.r.t all pair of entry weight* $(w_{Qs}, w_{Kr})$.

The proof of this theorem is somewhat trivial and similar to the second counterexample in Theorem B.2. Therefore, we will skip the full proof.

### B.2 Convexity of SFT

Even though a complete convex analysis is not possible due to the structure of the probability distribution function Prob, it is possible to prove the efficiency of this function during the training process via its pseudoconvexity, as all stationary points in a pseudoconvex are also global minimizers. The pseudoconvexity of the rational function class is verified by the following lemma.

**Lemma B.4.** *(Cambini & Martein, 2008) Let $z(x, c) = \frac{f(x)}{g(x)}$ be the ratio of two differentiable functions $f$ and $g$ defined on an open convex set $\mathcal{S} \subset \mathbb{R}^n$. If $f$ is convex and $g$ is positive and affine, then $z$ is a pseudoconvex function.*

Since ReLU is indifferentiable, the pseudoconvexity of Prob should be analyzed in separated orthants $\mathcal{O}_{\mathcal{I}}$ where $I \subset [n]$ is defined via the non-negativity of the input entries. Lemma B.4 directly implies that

**Theorem B.5.** *The probability distribution function* $\text{Prob} : \mathbb{R}^n \times \mathbb{R}^+ \to \mathbb{R}^n$ *defined as*

$$\text{Prob}(\mathbf{x}, c)[i] = \frac{\text{ReLU}(\mathbf{x}[i])}{\sum_k \text{ReLU}(\mathbf{x}[k]) + c},$$

*where $i \in [n]$ is a pseudoconvex function in orthant $\mathcal{O}_{\mathcal{I}}$ for all $\mathcal{I} \subset [n]$.*

*Proof.* Without loss of generality, consider the Prob function in the orthant $\mathcal{O}_1 = \mathcal{O}_{[n] \setminus \{1\}}$, which means that for any $\mathbf{x} \in \mathcal{O}_1$

$$\text{Prob}(\mathbf{x}, c)[i] = \begin{cases} 0 & \text{if } i = 1, \\ \dfrac{\mathbf{x}[i]}{\sum_k \mathbf{x}[k] + c} & \text{otherwise.} \end{cases}$$

From here, it is evident that the proof follows Lemma B.4 since $\mathbf{x}[k]$ is non-negative for all $k \neq 1$. Therefore, Prob is pseudoconvex in orthant $\mathcal{O}_1$ and, thus, in all orthants of $\mathbb{R}^n$. □

The probability distribution function Prob can only be pseudoconvex in each individual orthant $\mathcal{O}_{\mathcal{I}}$. However, due to the complete separation property, the stability during training is preserved. Consequently, we derive the pseudoconvexity power of SFT. But first, let us consider a modified version of SFT with linear activation in the FFN layer as follows:

**Theorem B.6.** *The SFT block in Algorithm* **??** *with FFN linear activation and no sampling has the following componentwise properties:*

- *Pseudoconvex with respect to $W_Q$, $B_Q$, $W_K$ and $B_K$;*

- *Pseudoconvex with respect to $W_{R_1}, B_{R_1}, W_{R_2}$ and $B_{R_2}$;*

- *Pseudoconvex with respect to $W_C$, $B_C$.*

*Proof.* Let $\mathbf{M} = SFT(W_Q, B_Q, W_K, B_K, W_{R_1}, W_{R_2}, W_C)$ be the output of a SFT block in Algorithm **??**.

- $\mathbf{M}$ is pseudoconvex with respect to $W_Q$, $W_K$, $B_Q$ and $B_K$.

  Consider the orthant $\mathcal{O}_{\mathbf{Q+K}} \cap \mathcal{O}_{\mathbf{Q-K}}$ of $\mathbf{Q}$ and $\mathbf{K}$, where $\mathcal{O}_{\mathbf{Q-K}}$ and $\mathcal{O}_{\mathbf{Q+K}}$ are the orthants in which the score function $\text{Score}(\mathbf{Q}, \mathbf{K})$ is differentiable and affine. Thus, it follows rom Theorem B.5 that each entry of the output of the Prob function can be written in the form $f(W_Q, W_K, B_Q, B_K)/(g(W_Q, W_K, B_Q, B_K) + C)$ where $f$ and $g$ are affine and $g$ is positive.

  As the remaining parts of SFT layer are linear transformations, they do not affect the affinity of $f$. Therefore, Lemma B.4 states the pseudoconvexity of the SFT block with respect to the inspected weights.

- $\mathbf{M}$ is pseudoconvex with respect to $W_{R_1}$, $W_{R_2}$, $B_{R_1}$ and $B_{R_2}$.

  This proof is similiar to which of $W_Q$ and $W_K$ and therefore is neglected.

- **M** is pseudoconvex with respect to $W_C$ and $B_C$. For the sake of simplity, Consider the $i^{th}$ entry of **C**, denoted as $\mathbf{C}[i] = \mathbf{X}[i]W_C + b_C$ where $\mathbf{1}_d b_C = B_C$, let there exist $W_{C1}$ and $W_{C2}$ such that:

$$\nabla\mathbf{M}(W_{C1})(W_{C2} - W_{C1}) \geq 0. \tag{15}$$

To prove the pseudoconvex dependency of **M** with respect to $W_C$ and $B_C$, consider the dependencies of $\mathbf{P}(W_C)$ with respect to $W_C$, and the pseudoconvexity w.r.t $b_C$ is analogous, where

$$\mathbf{P}(W_C) = SFT(\overline{W_Q}, \overline{B_Q}, \overline{W_K}, \overline{B_K}, \overline{W_{R_1}}, \overline{W_{R_2}}, W_C, \overline{b_C}),$$

and the overlined weights are treated as constants. This function can be written in a simpler form regarding each of its entry

$$\mathbf{P}(W_C)[i, j] = \frac{C_1}{C_2 + SP(\mathbf{X}[i]W_C + b_C)} + C_3,$$

where $C_1, C_2, C_3$ and $C_4$ are all constants, $C_2 > 0$ and SP denotes the softplus non-linearity. Returning to (15), we have:

$$-\frac{C_1}{(C_2 + SP(\mathbf{X}[i]W_{C1} + C_4))^2} SG(\mathbf{X}[i]W_{C1} + C_4)\mathbf{X}[i](W_{C2} - W_{C1}) \geq 0, \tag{16}$$

given that SG denotes the sigmoid function $SG(x) = 1/(1 + \exp(-x))$. Assume that $C_1 > 0$ (similar for $C_1 < 0$), then it follows from equation 16 that:

$$\mathbf{X}[i](W_{C2} - W_{C1}) \leq 0.$$

Thus for every $W_{C2}$ and $W_{C1}$ satisfying equation 15, we have:

$$\mathbf{P}(W_{C2}) \leq \mathbf{P}(W_{C1}).$$

By definition, **M** is pseudoconvex with respect to $W_C$ and $b_C$.

$\square$

However, in our proposed model, the FFN activation is GeLU. In order to show the pseudoconvexity of the our original SFT block, we analyze this via its quasiconvexity with the following theorem:

**Theorem B.7.** *(Ivanov, 2001) A function $f : \mathbb{R}^m \to \mathbb{R}^n$ that is quasiconvex is also pseudoconvex if the set of stationary points coincides with the set of global minimizers.*

From here, the pseudoconvexity of our model is proved.

**Theorem B.8.** *The SFT block with FFN GeLU activation and no sampling exhibits the same properties as in Theorem B.6.*

*Proof.* Let $\sigma(\cdot)$ be the shorthand notion of the GeLU non-linearity and $\mathbf{M}(\cdot)$ be the function of the linear activation FFN SFT block. Then, the function of our proposed SFT block is $\mathbf{G} := \sigma \circ \mathbf{M}$.

For the sake of simplicity, let $\mathbf{G}(\cdot)$ and $\mathbf{M}(\cdot)$ be a single-matrix-input and scalar-output function, which can represent any weight and entry respectively.

- Pseudoconvexity w.r.t the query, key, $\mathbf{R_1}$ and $\mathbf{R_2}$ scalers.

  Since the output of $\mathbf{M}(\cdot)$ is a fractional function with respect to these weights, as illustrated in the proof of Theorem B.6, it has no stationary points. Furthermore, due to the uniqueness of the global minimizer of $\sigma$, one can easily derive that $\mathbf{G}$, if quasiconvex, is also pseudoconvex by Theorem B.7.

Now we will prove the quasiconvexity of $\mathbf{G}(\cdot)$. Given that $\sigma$ is a quasiconvex scalar non-linearity, we consider the sublevel set of $\mathbf{G}$, defined as

$$
\begin{aligned}
L_{\mathbf{G}}(\alpha) &= \{\mathbf{W} \text{ s.t } \mathbf{G}(\mathbf{W}) \leq \alpha\} \\
&= \{\mathbf{W} \text{ s.t } u_1(\alpha) \leq \mathbf{M}(\mathbf{W}) \leq u_2(\alpha)\}.
\end{aligned}
$$

where $u_1(\cdot)$ and $u_2(\cdot)$ represent the lower and upper scalar bound of $\sigma$ with some level input.

However, as illustrated in Theorem B.6, $\mathbf{M}(\mathbf{W})$ is a fractional function with respect to $\mathbf{W}$ and so it is both pseudoconvex and pseudoconcave. Hence, $\mathbf{M}$ is also both quasiconvex and quasiconcave, which means that if we let

$$
L_{\mathbf{M}}^1(\alpha_1) = \{\mathbf{W} \text{ s.t } \mathbf{M}(\mathbf{W}) \geq \alpha_1\} \text{ and } L_{\mathbf{M}}^2(\alpha_2) = \{\mathbf{W} \text{ s.t } \mathbf{M}(\mathbf{W}) \leq \alpha_2\}
$$

be the superlevel and sublevel set of $\mathbf{M}(\mathbf{W})$ respectively, then they are convex sets for all $\alpha_1$ and $\alpha_2$. Therefore,

$$
L_{\mathbf{G}}(\alpha) = L_{\mathbf{M}}^1(u_1(\alpha)) \cap L_{\mathbf{M}}^2(u_2(\alpha))
$$

is also a convex set and the quasiconvexity of $\mathbf{G}(\mathbf{W})$ is derived.

- Pseudoconvexity w.r.t the leaky attention weights $W_C$ and $b_C$.

  Consider the function $\mathbf{P}(W_C)$ defined in Theorem B.6, by following the same proof steps, one can prove that $-\mathbf{P}(W_C)$ is also pseudoconvex. As a result, $\mathbf{P}(W_C)$ is pseudoconcave.

  It is also evident that $\mathbf{P}(W_C)$ has no stationary points. Hence, by replicating the proof of the other weights, it is sufficient to show that the SFT block is also pseudoconvex with respect to $W_C$, and also to $b_C$.

$\square$

## B.3 Gradient Analysis of SFT

To theoretically analyze the effectiveness of SFT against flat and sharp points, we evaluate the local adaptive lower bound and a global upper bound for the gradient norm squared of the weights.

**Theorem B.9.** *Let $\mathbf{Sa} = \mathbf{Sa}(W_Q, W_K, W_V) = \text{Prob}(\text{Score}(\mathbf{X}W_Q, \mathbf{X}W_K), \mathbf{C})\mathbf{X}W_V + \alpha\mathbf{X}$ be the SFT self-attention layer output with the simplication of $\mathbf{C}$ being treated as a positive constant matrix. Assume that all weights are initialized via He initialization (He et al., 2015) where each entry independently follows the normal distribution $\mathcal{N}(0, \sigma^2)$ and let $W_D = W_Q - W_K$, then for any positive matrices $W_{QQ}^L, W_{QQ}^U, W_{KK}^L, W_{KK}^U, W_{DD}^L, W_{DD}^U$ such that $W_{QQ}^L < W_{QQ}^U$, $W_{KK}^L < W_{KK}^U$, $W_{DD}^L < W_{DD}^U$ and $W_Q W_Q^\top \in [W_{QQ}^L, W_{QQ}^U]$, $W_K W_K^\top \in [W_{KK}^L, W_{KK}^U]$ and $W_D W_D^\top \in [W_{DD}^L, W_{DD}^U]$ there exist two positive scalar-valued positive function $f_1, f_2$ and positive constants $C_Q, C_V$ such that:*

$$
\mathbb{E}\left\|\frac{\partial \mathbf{Sa}}{\partial W_V}\right\|_F^2 \in \left(f_V(W_{QQ}^L, W_{QQ}^U, W_{KK}^L, W_{KK}^U), C_V\right),
$$

$$
\mathbb{E}\left\|\frac{\partial \mathbf{Sa}}{\partial W_Q}\right\|_F^2 \in \left(f_Q(W_{DD}^L, W_{DD}^U, W_{KK}^L, W_{KK}^U), C_Q\right).
$$

*and*

$$
f_V(W_{QQ}^L, W_{QQ}^U, W_{KK}^L, W_{KK}^U) = \Omega(n), \qquad C_V = O(n),
$$

$$
f_Q(W_{DD}^L, W_{DD}^U, W_{KK}^L, W_{KK}^U) = \Omega(1), \qquad C_Q = O(1).
$$

*Proof.* First, let us note some remarks on matrix calculus and properties of the Kronecker product $\otimes$, shown at (Magnus & Neudecker, 1999) (Singh et al., 2021). Given some matrices $\mathbf{A} \in \mathbb{R}^{s_1 \times s_2}, \mathbf{B} \in \mathbb{R}^{s_2 \times s_3}, \mathbf{C} \in \mathbb{R}^{s_3 \times s_4}, \mathbf{D} \in \mathbb{R}^{s_4 \times s_5}$ and a matrix variable $\mathbf{W} \in \mathbb{R}^{s_2 \times s_2}$, then:

$$\frac{\partial \mathbf{AWB}}{\partial \mathbf{W}} = \mathbf{A} \otimes \mathbf{B}^\top.$$

Additionally, we also mention some useful properties regarding the trace operator, denoted as $\text{tr}(\cdot)$,

$$\text{tr}(\mathbf{A} \otimes \mathbf{B}) = \text{tr}(\mathbf{A})\,\text{tr}(\mathbf{B}),$$
$$(\mathbf{AC}) \otimes (\mathbf{BD}) = (\mathbf{A} \otimes \mathbf{B})(\mathbf{C} \otimes \mathbf{D}).$$

We would also like to recall to the Jensen inequality for the expectation of convex and concave function. Let $\mathbb{E}$ denotes the expectation operator, and $X$ be a random variable. Then, for any scalar function $\phi : \mathbb{R} \to \mathbb{R}$,

- $\mathbb{E}(\phi(X)) \geq \phi(\mathbb{E}(X))$ if $\phi$ is convex,

- $\mathbb{E}(\phi(X)) \leq \phi(\mathbb{E}(X))$ if $\phi$ is concave.

Now we are ready for the proof, let us consider the gradient norm w.r.t $W_V$. For simplicity, we have the notations

$$\mathbf{T}_i = \text{Score}(\mathbf{X}[i]W_Q, \mathbf{X}W_K),$$
$$\mathbf{R}_i = \text{Prob}(\mathbf{T}_i).$$

Note that

$$\left\| \frac{\partial \mathbf{Sa}}{\partial W_V} \right\|_F^2 = \sum_{i=1}^n \left\| \frac{\partial \mathbf{Sa}[i]}{\partial W_V} \right\|_F^2$$

$$= \sum_{i=1}^n \| \mathbf{R}_i \mathbf{X} \otimes \mathbf{I}_d \|_F^2$$

$$= \sum_{i=1}^n \text{tr}\left( ((\mathbf{R}_i X) \otimes \mathbf{I}_d)\left( (X^\top \mathbf{R}_i^\top) \otimes \mathbf{I}_d \right) \right)$$

$$= \sum_{i=1}^n \text{tr}\left( (\mathbf{R}_i \mathbf{X}\mathbf{X}^\top \mathbf{R}_i^\top) \otimes \mathbf{I}_d \right)$$

$$= \sum_{i=1}^n d\, \text{tr}\left( \mathbf{R}_i \mathbf{X}\mathbf{X}^\top \mathbf{R}_i^\top \right)$$

$$= \sum_{i=1}^n d\, \| \mathbf{R}_i \mathbf{X} \|_F^2$$

$$= \sum_{i=1}^n \sum_{j=1}^n d\, \| \mathbf{R}_i[j] \mathbf{X}[j] \|_F^2 .$$

However, we have:

$$\mathbf{R}_i[j] = \frac{\text{ReLU}\left( \max(\mathbf{Q}[i], \mathbf{K}[j]) \right)}{\sum_{k=1}^n \text{ReLU}\left( \max(\mathbf{Q}[i], \mathbf{K}[k]) \right) + \mathbf{C}[i]}.$$

In order to linearize the max operator, let $e_{ij} \in \{0, 1\}$ be the query-key indicator such that:

$$\max(\mathbf{Q}[i], \mathbf{K}[j]) = e_{ij}\mathbf{Q}[i] + (1 - e_{ij})\mathbf{K}[j].$$

Similiar to the $e_{ij}$, let $d_{ij} \in \{0,1\}$ represents the ReLU non-linearity by having

$$\text{ReLU}\left(\max(\mathbf{Q}[i], \mathbf{K}[j])\right) = d_{ij} \max(\mathbf{Q}[i], \mathbf{K}[j]) = d_{ij}\left(e_{ij}\mathbf{Q}[i] + (1 - e_{ij})\mathbf{K}[j]\right).$$

By this formulation, we have

$$
\left\| \frac{\partial \mathbf{Sa}}{\partial W_V} \right\|_F^2 = d \sum_{i=1}^{n} \sum_{j=1}^{n} \frac{d_{ij}e_{ij}\mathbf{Q}[i]\mathbf{Q}^\top[i] + d_{ij}(1 - e_{ij})\mathbf{K}[j]\mathbf{K}^\top[j]}{\left(\sum_{k=1}^{n} d_{ik}e_{ik}\mathbf{Q}[i] + d_{ik}(1 - e_{ik})\mathbf{K}[k] + \mathbf{C}[i]\right)^2} \|\mathbf{X}[j]\|_F^2
$$

$$
= d \sum_{i=1}^{n} \sum_{j=1}^{n} \frac{d_{ij}e_{ij}\operatorname{tr}(\mathbf{X}^\top[i]\mathbf{X}[i]W_Q W_Q^\top) + d_{ij}(1 - e_{ij})\operatorname{tr}(\mathbf{X}^\top[j]\mathbf{X}[j]W_K W_K^\top)}{\left(\sum_{k=1}^{n} d_{ik}e_{ik}\operatorname{tr}(\mathbf{X}[i]W_Q) + d_{ik}(1 - e_{ik})\operatorname{tr}(\mathbf{X}[i]W_K) + \mathbf{C}[i]\right)^2} \|\mathbf{X}[j]\|_F^2
$$

$$
\geq \frac{d}{3n} \sum_{i=1}^{n} \sum_{j=1}^{n} \frac{d_{ij}e_{ij}\operatorname{tr}(\mathbf{X}^\top[i]\mathbf{X}[i]W_Q W_Q^\top) + d_{ij}(1 - e_{ij})\operatorname{tr}(\mathbf{X}^\top[j]\mathbf{X}[j]W_K W_K^\top)}{\sum_{k=1}^{n} d_{ik}e_{ik}\operatorname{tr}(\mathbf{X}^\top[i]\mathbf{X}[i]W_Q W_Q^\top) + d_{ik}(1 - e_{ik})\operatorname{tr}(\mathbf{X}^\top[k]\mathbf{X}[k]W_K W_K^\top) + \mathbf{C}[i]^2} \|\mathbf{X}[j]\|_F^2.
$$

To express this more concisely, we have following shorthand notations:

$$\mathbf{A}_{Qi} = \frac{d}{3n} \sum_{j=1}^{n} d_{ij}e_{ij} \|\mathbf{X}[j]\|_F^2 \, \mathbf{X}^\top[i]\mathbf{X}[i],$$

$$\mathbf{A}_{Ki} = \frac{d}{3n} \sum_{j=1}^{n} d_{ij}(1 - e_{ij}) \|\mathbf{X}[j]\|_F^2 \, \mathbf{X}^\top[j]\mathbf{X}[j],$$

$$\mathbf{B}_{Qi} = \sum_{k=1}^{n} d_{ik}e_{ik}\mathbf{X}^\top[i]\mathbf{X}[i],$$

$$\mathbf{B}_{Ki} = \sum_{k=1}^{n} d_{ik}(1 - e_{ik})\mathbf{X}^\top[k]\mathbf{X}[k].$$

By doing this, we derive that:

$$\left\| \frac{\partial \mathbf{Sa}}{\partial W_V} \right\|_F^2 \geq \sum_{i=1}^{n} \frac{\operatorname{tr}(\mathbf{A}_{Qi}W_{QQ}) + \operatorname{tr}(\mathbf{A}_{Ki}W_{KK})}{\operatorname{tr}(\mathbf{B}_{Qi}W_{QQ}) + \operatorname{tr}(\mathbf{B}_{Ki}W_{KK}) + \mathbf{C}[i]^2},$$

where $W_{QQ} = W_Q W_Q^\top$ and $W_{KK} = W_K W_K^\top$.

We delve into the concept of convex envelope, which seeks the highest convex underestimator of a given function. Additionally, the local minina of the convex envelope is also the global minima of the primal function.

However, in this proof, we only focus on the convex properties. Specifically, consider the non-convex lower bound function

$$\mathbf{LB}_i(W_{QQ}, W_{KK}) = \frac{\operatorname{tr}(\mathbf{A}_{Qi}W_{QQ}) + \operatorname{tr}(\mathbf{A}_{Ki}W_{KK})}{\operatorname{tr}(\mathbf{B}_{Qi}W_{QQ}) + \operatorname{tr}(\mathbf{B}_{Ki}W_{KK}) + \mathbf{C}[i]^2}, \tag{17}$$

and based on the convex envelope of the bivariate rational function discovered at (Tawarmalani & Sahinidis, 2002), we can convexify the multivariate rational function $\mathbf{LB}_i$ by considering the convexity with respect to each entry one at a time. For convenience, we let $\mathbf{Af}_i(W_{QQ}, W_{KK})$ be the affine function denominator of equation 17. As a result, we can derive a componentwise convex lower bound of $\mathbf{LB}_i$ called $\mathbf{CCLB}_i$, which can be expressed in the form

$$\mathbf{CCLB}_i(W_{QQ}, W_{KK}) = \frac{r_i}{\mathbf{Af}_i(W_{QQ}, W_{KK})} \prod_{s,k,p,q=1}^{d} \mathbf{Cci}_{Qsr}\left(\frac{\left(w_{Qsr} + \sqrt{w_{Qsr}^L w_{Qsr}^U}\right)^2}{\left(\sqrt{w_{Qsr}^L} + \sqrt{w_{Qsr}^U}\right)^2}\right) \mathbf{Cci}_{Kpq}\left(\frac{\left(w_{Kpq} + \sqrt{w_{Kpq}^L w_{Kpq}^U}\right)^2}{\left(\sqrt{w_{Kpq}^L} + \sqrt{w_{Kpq}^U}\right)^2}\right)$$

where:

- $w_{Qsr}, w_{Kpq}$ is the $[s, r]$ and $[p, q]$ location entry of $W_{QQ}$ and $W_{KK}$ respectively,

- $w_{Qsr}^L$ and $w_{Qsr}^U$ are the lower and upper bound of $w_{Qsr}$,

- $w_{Kpq}^L$ and $w_{Kpq}^U$ are the lower and upper bound of $w_{Kpq}$,

- $r_i$ is a constant and is independent of the bounds,

- $\mathbf{Cci}_{Qsr}(\cdot)$ is the identity function if $w_{Qsr}$ is concave in the $\mathbf{LB}_i$ function, and 1 otherwise,

- $\mathbf{Cci}_{Kpq}(\cdot)$ is defined similar to $\mathbf{Cci}_{Qsr}(\cdot)$.

With this formulation, we take the expectation of the gradient norm square and use the Jensen inequality, which leads to:

$$\mathbb{E} \left\| \frac{\partial \mathbf{Sa}}{\partial W_V} \right\|_F^2 \geq \sum_{i=1}^n \frac{r_i}{\mathbf{Af}_i(\sigma^2 \mathbf{I}_n, \sigma^2 \mathbf{I}_n)} \prod_{s,r,p,q=1}^d \mathbf{Cci}_{Qsr} \left( \frac{\left( \sigma^2 \delta_{sr} + \sqrt{w_{Qsr}^L w_{Qsr}^U} \right)^2}{\left( \sqrt{w_{Qsr}^L} + \sqrt{w_{Qsr}^U} \right)^2} \right) \mathbf{Cci}_{Kpq} \left( \frac{\left( \sigma^2 \delta_{pq} + \sqrt{w_{Kpq}^L w_{Kpq}^U} \right)^2}{\left( \sqrt{w_{Kpq}^L} + \sqrt{w_{Kpq}^U} \right)^2} \right)$$

$$\text{(18)}$$

$$\triangleq f_V(W_{QQ}^L, W_{QQ}^U, W_{KK}^L, W_{KK}^U) = \Omega(n), \tag{19}$$

where $\delta_{sr} = 1$ if $s = r$ and 0 otherwise.

Now let us consider the gradient norm with respect to the query:

$$\left\| \frac{\partial \mathbf{Sa}}{\partial W_Q} \right\|_F^2 = \sum_{i=1}^n \left\| \frac{\partial \mathbf{Sa}[i]}{\partial W_Q} \right\|_F^2.$$

Consider the gradient norm for each $i \in [n]$, by applying the chain rule we have:

$$\left\| \frac{\partial \mathbf{Sa}[i]}{\partial W_Q} \right\|_F^2 = \left\| \frac{\partial \mathbf{Sa}[i]}{\partial \mathbf{R}_i} \frac{\partial \mathbf{R}_i}{\partial \mathbf{T}_i} \frac{\partial \mathbf{T}_i}{\partial \mathbf{Q}[i]} \frac{\partial \mathbf{Q}[i]}{\partial W_Q} \right\|_F^2$$

$$= \left\| W_V^\top \mathbf{X}^\top \frac{\partial \mathbf{R}_i}{\partial \mathbf{T}_i} \mathbf{e}_i \mathbf{X}[i] \right\|_F^2,$$

where $\mathbf{e}_i = (e_{ij})_{j \in [n]}$ is the query-key binary gate vector. Continuing the formulation,

$$\mathbb{E} \left\| \frac{\partial \mathbf{Sa}[i]}{\partial W_Q} \right\|_F^2 = \mathbb{E} \left( \mathrm{tr} \left( W_V W_V^\top \mathbf{X}^\top \frac{\partial \mathbf{R}_i}{\partial \mathbf{T}_i} \mathbf{e}_i \mathbf{X}[i] \mathbf{X}^\top [i] \mathbf{e}_i^\top \frac{\partial \mathbf{R}_i^\top}{\partial \mathbf{T}_i^\top} \mathbf{X} \right) \right)$$

$$= \sigma^2 \|\mathbf{X}[i]\|_F^2 \, \mathrm{tr} \left( \mathbf{X} \mathbf{X}^\top \mathbb{E} \left( \frac{\partial \mathbf{R}_i}{\partial \mathbf{T}_i} \mathbf{e}_i \mathbf{e}_i^\top \frac{\partial \mathbf{R}_i^\top}{\partial \mathbf{T}_i^\top} \right) \right)$$

$$= \sigma^2 \|\mathbf{X}[i]\|_F^2 \, \mathbb{E} \left\| \mathbf{X}^\top \frac{\partial \mathbf{R}_i}{\partial \mathbf{T}_i} \mathbf{e}_i \right\|_F^2$$

$$= \sum_{k=1}^d \sigma^2 \|\mathbf{X}[i]\|_F^2 \, \mathbb{E} \left\| \mathbf{X}[:, k]^\top \frac{\partial \mathbf{R}_i}{\partial \mathbf{T}_i} \mathbf{e}_i \right\|_F^2.$$

Let $W_D = W_Q - W_K$ and $\mathbf{Z}_i = \dfrac{\partial \mathbf{R}_i}{\partial \mathbf{T}_i} \mathbf{e}_i$, we have:

$$\mathbf{Z}_i[j] = \frac{d_{ij}e_{ij}\left(\sum_{\substack{k=1 \\ k \neq j}}^n d_{ik}\mathbf{T}_i[k] - d_{ik}\mathbf{T}_i[j]\right)}{(d_{ik}\mathbf{T}_i[k] + \mathbf{C}[i])^2}.$$

And since $\mathbf{T}_i[k] = e_{ik}\mathbf{Q}[i] + (1 - e_{ik})\mathbf{K}[k] = e_{ik}\mathbf{X}[i]W_D + ((1-e_{ik})\mathbf{X}[k] + e_{ik}\mathbf{X}[i])W_K$, we derive that:

$$\left\|\mathbf{X}[:,k]^\top \mathbf{Z}_i\right\|_F^2 = \frac{\sum_{j=1}^n \mathbf{X}[j,k]^2 d_{ij}e_{ij}\left(\sum_{\substack{k=1 \\ k \neq j}}^n d_{ik}(1-e_{ik})\right)^2 (\mathbf{X}[i]W_D)^2}{\left(\sum_{k=1}^n d_{ik}e_{ik}\mathbf{X}[i]W_D + d_{ik}((1-e_{ik})\mathbf{X}[k] + e_{ik}\mathbf{X}[i])W_K + \mathbf{C}[i]\right)^4}$$

$$\geq \frac{1}{9n^2} \frac{\sum_{j=1}^n \mathbf{X}[j,k]^2 d_{ij}e_{ij}\left(\sum_{\substack{k=1 \\ k \neq j}}^n d_{ik}(1-e_{ik})\right)^2 (\mathbf{X}[i]W_D)^2}{\left(\sum_{k=1}^n d_{ik}e_{ik}(\mathbf{X}[i]W_D)^2 + d_{ik}(((1-e_{ik})\mathbf{X}[k] + e_{ik}\mathbf{X}[i])W_K)^2 + \mathbf{C}[i]^2\right)^2}.$$

Now let $W_{DD} = W_D W_D^\top$ and confine $W_{DD}$ in a hypercube $(W_{DD}^L, W_{DD}^U)$ similarly to $W_{QQ}$ and $W_{KK}$, we have

$$(\mathbf{X}[i]W_D)^2 = \text{tr}(\mathbf{X}[i]^\top \mathbf{X}[i]W_D W_D^\top) \leq \|\mathbf{X}[i]\|_F^2 \|W_{DD}\|_F \leq \|\mathbf{X}[i]\|_F^2 C\left(W_{DD}^L, W_{DD}^U\right),$$

where $C(\cdot, \cdot)$ is some scalar-valued positive function dependent only on the bounds of $W_{DD}$.

By applying similar evaluations for $W_{KK}$ and following the techniques used for the analysis $W_V$ gradient norm, such that it is sufficient to deduce that for some constant matrices $A_{Di}, B_{Di}$ and $B_{Ki}$ that are bound-independent, we have:

$$\left\|\mathbf{X}[:,k]^\top \mathbf{Z}_i\right\|_F^2 \geq C\left(W_{DD}^L, W_{DD}^U, W_{KK}^L, W_{KK}^U\right) \frac{\text{tr}(A_{Di}W_{DD})}{\text{tr}(B_{Di}W_{DD}) + \text{tr}(B_{Ki}W_{KK}) + \mathbf{C}[i]^2}.$$

It is evident of the proof from this point as it is analogous to the previous part. Therefore, we deduce another adaptive lower bound for the gradient norm of $W_Q$ as there exist a non-negative scalar function $f_Q$ such that:

$$\mathbb{E}\left\|\frac{\partial \mathbf{Sa}}{\partial W_Q}\right\|_F^2 \geq f_Q\left(W_{DD}^L, W_{DD}^U, W_{KK}^L, W_{KK}^U\right) = \Omega(1). \tag{20}$$

Regarding the upper bound, we will achieve this by acquiring a sufficiently large universal constant. Simply by the norm product inequality, we have:

$$\mathbb{E}\left\|\frac{\partial \mathbf{Sa}}{\partial W_V}\right\|_F^2 \leq d \max_i \|\mathbf{X}[i]\|_F^2 \sum_{i=1}^n \|\mathbf{R}_i\|_F^2 \leq dn \max_i \|\mathbf{X}[i]\|_F^2 \triangleq C_V = O(n). \tag{21}$$

Combining Eq 19 and Eq 21, the complexity of $\mathbb{E}\left\|\dfrac{\partial \mathbf{Sa}}{\partial W_V}\right\|_F^2$ is derived to be $\Theta(n)$.

Also,

$$\mathbb{E}\left\|\frac{\partial \mathbf{Sa}}{\partial W_Q}\right\|_F^2 \leq \sum_{i=1}^n \sum_{k=1}^d \sigma^2 \|\mathbf{X}[i]\|_F^2 \mathbb{E}\left\|\mathbf{X}[:,k]^\top \mathbf{Z}_i\right\|_F^2.$$

Since $W_D = W_Q - W_K$, we get $\mathrm{Var}(W_D) = \mathrm{Var}(W_Q) + \mathrm{Var}(W_K) = 2\sigma^2 \mathbf{I}_n$. Alongside this, notice that the indicators $d_{ij}$ where $i, j \in [n]$ can be treated as binary gates that filter out negative values, therefore:

$$\mathbb{E}\left\|\mathbf{X}[:,k]^\top \mathbf{Z}_i\right\|_F^2 \leq \mathbb{E}\frac{\sum_{j=1}^n \mathbf{X}[j,k]^2 d_{ij} e_{ij} \left(\sum_{\substack{k=1 \\ k\neq j}}^n d_{ik}(1-e_{ik})\right)^2 (\mathbf{X}[i]W_D)^2}{\left(\sum_{k=1}^n d_{ik}e_{ik}\mathbf{X}[i]W_D + \mathbf{C}[i]\right)^4}$$

$$\leq \mathbb{E}\frac{\sum_{j=1}^n \mathbf{X}[j,k]^2 d_{ij} e_{ij} \left(\sum_{\substack{k=1 \\ k\neq j}}^n d_{ik}(1-e_{ik})\right)^2}{\left(\sum_{k=1}^n d_{ik}e_{ik}\mathbf{X}[i]W_D + \mathbf{C}[i]\right)^2 \left(\frac{\mathbf{C}[i]}{\mathbf{X}[i]W_D} + \sum_{k=1}^n d_{ik}e_{ik}\right)^2}$$

$$\leq \mathbb{E}\frac{\sum_{j=1}^n \mathbf{X}[j,k]^2 d_{ij} e_{ij} \left(\sum_{\substack{k=1 \\ k\neq j}}^n d_{ik}(1-e_{ik})\right)^2}{4\mathbf{C}[i]^2 \left(\sum_{k=1}^n d_{ik}e_{ik}\right)^4}.$$

Finally, the global upper bound for the gradient norm squared of $W_V$ is concluded:

$$\mathbb{E}\left\|\frac{\partial \mathbf{Sa}}{\partial W_Q}\right\|_F^2 \leq \sum_{i=1}^n \sum_{k=1}^d \sigma^2 \|\mathbf{X}[i]\|_F^2 \mathbb{E}\left(\frac{\sum_{j=1}^n \mathbf{X}[j,k]^2 d_{ij} e_{ij} \left(\sum_{\substack{k=1 \\ k\neq j}}^n d_{ik}(1-e_{ik})\right)^2}{4\mathbf{C}[i]^2 \left(\sum_{k=1}^n d_{ik}e_{ik}\right)^4}\right) \triangleq C_Q = O(1). \quad (22)$$

From Eq. 20 and Eq. 22, we conclude that the complexity of $\mathbb{E}\left\|\frac{\partial \mathbf{Sa}}{\partial W_Q}\right\|_F^2$ is $\Theta(1)$. $\qquad\square$

## C    Datasets

The ModelNet40 dataset (Wu et al., 2015) consists of 12,311 pre-aligned shapes divided into 40 classes, where the train and test sets consist of 9,843 instances and 2,468 instances respectively. ModelNet40 is the pioneer large-scale 3D CAD dataset. Unlike previous CAD datasets (Shilane et al., 2004), ModelNet40 is the pioneer large-scale dataset that is diverse in terms of both class and samples per class.

The ShapeNetPart dataset consists of 16,881 pre-aligned 3D shapes from 16 categories and is a part of a larger dataset: ShapeNetCore (51,300 3D models) (Chang et al., 2015). The 3D shapes in the ShapeNetPart dataset are annotated with 50 segmentation parts in total representing virtual real-world 3D semantic models. ShapeNet provides a diverse variety of shape annotations and corresponding shapes. The full ShapeNet

dataset a is multitude containing upright and front orientation vectors, parts and keypoints, shape symmetries, but we only account for the part-segmenting task in our work.

The Long Range Arena Benchmark (LRA) (Tay et al., 2021) is a composition of 5 tasks: ListOps (Nangia & Bowman, 2018), ImDB review (Maas et al., 2011), ACL Anthology Network (Radev et al., 2009), Grayscaled CIFAR-10 (Krizhevsky & Hinton, 2009), and Pathfinder (Linsley et al., 2018). These five tasks are all classification and they feature very long sequences: ListOps (2,048 tokens), ImDB review (1,024 tokens), ACL Anthology Network (4,096 tokens), Grayscaled CIFAR-10 (1,024 tokens), and Pathfinder (1,024 tokens). All five tasks involve tackling long-ranged data structures and manifold categorizations, challenging networks's generalization powers as well as memory and time efficiencies.

We used three datasets in Long Range Graph Benchmark (LRGB) (Dwivedi et al., 2022b): Peptides-func (Singh et al., 2015), Peptides-struct (Singh et al., 2015), and PascalVOC-sp (Everingham et al., 2010). Peptides-func is a Graph Multiclass-Classification task, that features graphs with an averaged number of nodes of 150. Similarly, Peptides-struct is a Graph-Regression task, with the same averaged number of nodes. PascalVOC-sp is a Node-Classification task, with an averaged number of nodes of 479. While the Peptide datasets measure the long-range relationship deduction of models, the PascalVOC-sp dataset measures local pattern recognition through the Node-Classification tasks. This combination provides a good enough evaluation of efficient transformers' performance in the Graph data modality.

## D   Reproducibility

This section is devoted to providing information to assist the reproducibility of our work and explain the technicality of our experimental code.

In our work, we conducted the following experiments:

- Experiment to measure the performance of our model in Object-Classification on ModelNet40 (1.1)

- Experiment to measure the performance of our model in Semantic-Segmentation on ShapeNetPart (1.2)

- Experiment to measure the performance of our model in Object-Classification on ModelNet40 under rotational invariant constraint (1.3)

- Experiment to measure the performance of our model in Semantic-Segmentation on ShapeNetPart under rotational invariant constraint (1.4)

- Experiment to measure the performance of our model in Graph-Multiclass-Classification on Peptide-func (2.1)

- Experiment to measure the performance of our model in Graph-Regression on Peptide-struct (2.2)

- Experiment to measure the performance of our model in Graph-Node-Classification on PascalVOC-sp (2.3)

- Experiment to measure the performance of our model in LRA-Benchmark-ListOPS (3.1)

- Experiment to measure the performance of our model in LRA-Benchmark-Text (3.2)

- Experiment to measure the performance of our model in LRA-Benchmark-Retrieval (3.3)

- Experiment to measure the performance of our model in LRA-Benchmark-Image (3.4)

- Experiment to measure the performance of our model in LRA-Benchmark-Pathfinder (3.5)

- Experiment to show the effectiveness of our attention formulation in ModelNet40 (4.1)

- Experiment to analyze the performance-efficiency tradeoff in ModelNet40 (4.2)

- Experiment to measure model efficiency (4.3)

- Experiment to investigate rank injection phenomena (5)

### D.1 Experiment details

This subsection provides information on the experiment settings of both our main experiments and our ablation study experiments.

All the experiments on ModelNet40 (1.1, 1.3, 4.1, 4.2) use uniform point cloud sampling technique to sample 1024 points as input tokens. Experiments on ShapeNetPart do not use uniform sampling technique; it randomly samples 2500 points.

All the point cloud experiments (1.1, 1.2, 1.3, 1.4, 4.1, 4.2) under rotational invariant constraint use both point cloud coordinates and surface normal vectors. Under rotational invariant constraint, the input token value are replaced by tensor filled with one, and the relative positional embedding matrix is a Squared Euclidean Distance Matrix from point coordinates concatenated with Pairwise Dot-Product of surface normal vectors; these two features are rotational invariant.

All experiments on LRG-Benchmark (2.1, 2.2, 2.3) use graph adjacency matrix as relative information between tokens. The subscripted adjacency matrices are fed into each layer based on the output of the sampling module.

All experiments on LRA-Benchmark (3.1, 3.2, 3.3, 3.4, 3.5) use a learnable linear combination of Sinusoid1D as relative positional encoding; the sinusoid vectors are linearly combined, then a pairwise hadamard product is used to construct relative matrix between tokens. This method uses a sinusoidal matrix $\mathbf{S}$ having shape $n \times d$, defined as follows:

$$\mathbf{S}[i, 2j] = \sin(\frac{i}{10000^{\frac{j}{d}}}), \tag{23}$$

$$\mathbf{S}[i, 2j + 1] = \cos(\frac{i}{10000^{\frac{j}{d}}}), \tag{24}$$

$n, d$ are number of tokens and number of dimensions, respectively. The sinusoid matrix fed into the sampler, resulting in matrices of sampled positional tokens $\mathbf{S}_{\text{top}}, \mathbf{S}_{\text{rand}}$. Relative position matrice is constructed by pairwise hadamard product, which results in tensor $\mathbf{A}_{\text{top}}$ and $\mathbf{A}_{\text{rand}}$ of shape $(n, n', d)$:

$$\mathbf{A}[i, j, k] = \mathbf{S}[i, k] * \mathbf{S}[j, k] \tag{25}$$

A learnable linear layer is then applied to $\mathbf{A}$, transforming it into a tensor of shape $(n, n', h)$, where $h$ is the number of attention heads. The remaining steps regarding relative positional encoding can be traced back in Section 4.3, as we have described the method to generate $\mathbf{X_R}$.

### D.2 Implementation details

This subsection provides information on how we organize and implement our experimental software.

All the experiments involving model training use TorchCompile to compile the entire model into one optimized kernel. This reduces around 40% of the computational cost of our models.

Our implementation for graph data modality is not yet able to support very large graphs due to the usage of subscripted adjacency matrices. This would be fixed in the future if we implement direct sparse operator converting edge list into subscripted adjacency matrices, ensuring linear runtime.

Our code is split into three modules:

- Data-related Modules: data loaders and data augmentation modules for all of our experimented tasks.

- Layers: code for our differentiable sampling-without-replacement module, SFT-layer with various configurations, and task-head. The code for our differentiable sampling-without-replacement module is separated from other codes; therefore, it is possible to use our sampling module to more optimized transformer implementation, such as the PyTorch one.

- Models: methods to parse point cloud models, sequential models, and graph models.

The experiments in which we calculate Flops of submodules (4.3) use the Calflop Python library. In this experiment, we disable TorchCompile.

### D.3 Training Strategies

This subsection provides information on how we train our models across tasks; including the choice of optimizers, optimizer hyperparameters, learning rate schedulers, and data augmentations.

All of our model uses AdamW optimizer, StepLR learning rate scheduler, max gradient norm clipping, and Untuned Linear Learning-Rate Warmup (Ma & Yarats, 2021). Therefore, the training-strategy-related hyperparameters are:

- Optimizer
  - Number of Epochs
  - Batch Size
  - Learning Rate
  - L2-Weight-Decay

- Learning Rate Scheduler
  - Step Size
  - Decay Rate

- Max Gradient Norm Clipping
  - Max Gradient Norm

Point Cloud tasks use data preprocessing and data augmentation techniques as the following:

- Data preprocessing: All of the point clouds are centralized

- Data preprocessing: The point cloud size is normalized by dividing the point coordinates by the distance from point cloud centroid and the farthest point to the centroid

- Data augmentation: Randomly change the size of point clouds by multiplying point cloud coordinates with a uniformly sampled scalar in range of [0.6, 1.4]

Peptide tasks use data preprocessing as the following:

- Data preprocessing: Node features go through a 17-class one-hot vectorization

- Data preprocessing: Edge features are concatenated with 1-valued tensor

Here, we list the training hyperparameter for each experiment in Table 8.

### D.4 Architectural Specification

This subsection provides information on the architectural hyperparameters of models across tasks.

All of the models we experimented use a feedforward network expansion rate of 4 (this implies the FFN module in SFT transforms d-dimensional tokens into 4d-dimensional tokens then back to d-dimensional tokens).

To combat overfitting, we have a slight difference compared to the vanilla transformer: we introduce dropout layers with different dropout rates: a dropout layer for Q and K vectors is called attention score dropout, a

Table 8: Training Hyperparameters of Experiments

| Experiment | 1.1 | 1.2 | 1.3 | 1.4 | 2.1 | 2.2 | 2.3 | 3.1 | 3.2 | 3.3 | 3.4 | 3.5 | 4.1 | 4.2 | 4.3 | 5 |
|---|---|---|---|---|---|---|---|---|---|---|---|---|---|---|---|---|
| Number of Epoch | 600 | 600 | 1000 | 1000 | 600 | 600 | 600 | 50 | 200 | 80 | 50 | 200 | 600 | 600 | 600 | 1000 |
| Batch Size | 32 | 32 | 32 | 32 | 32 | 32 | 32 | 64 | 64 | 32 | 64 | 64 | 32 | 32 | 32 | 32 |
| Learning Rate | 0.001 | 0.001 | 0.001 | 0.001 | 0.0008 | 0.0003 | 0.001 | 0.001 | 0.001 | 0.001 | 0.001 | 0.001 | 0.001 | 0.001 | 0.001 | 0.001 |
| L2 Weight Decay | 0.1 | 0.1 | 0.1 | 0.1 | 0.17 | 0.12 | 0.1 | 0.01 | 0.01 | 0.01 | 0.02 | 0.01 | 0.1 | 0.1 | 0.1 | 0.1 |
| Step Size | 30 | 30 | 30 | 30 | 30 | 30 | 30 | 20 | 20 | 20 | 20 | 20 | 30 | 30 | 30 | 30 |
| Decay Rate | 0.8 | 0.8 | 0.8 | 0.8 | 0.8 | 0.8 | 0.9 | 0.9 | 0.9 | 0.9 | 0.9 | 0.9 | 0.8 | 0.8 | 0.8 | 0.8 |
| Max Gradient Norm | 1.0 | 1.0 | 1.0 | 1.0 | 16.0 | 16.0 | 1.0 | 1.0 | 1.0 | 1.0 | 1.0 | 1.0 | 1.0 | 1.0 | 1.0 | 1.0 |

Table 9: Architectural Hyperparameters of Experiments

| Experiment | 1.1 | 1.2 | 1.3 | 1.4 | 2.1 | 2.2 | 2.3 | 3.1 | 3.2 | 3.3 | 3.4 | 3.5 |
|---|---|---|---|---|---|---|---|---|---|---|---|---|
| Number of Sampled Tokens | 256 | 256 | 256 | 256 | 75 | 75 | 150 | 256 | 256 | 256 | 256 | 256 |
| Drop Token Probability | 0.5 | 0.2 | 0.5 | 0.2 | 0.3 | 0.3 | 0.3 | 0.0 | 0.3 | 0.0 | 0.3 | 0.0 |
| Token Embedding Size | 256 | 256 | 256 | 256 | 144 | 144 | 160 | 128 | 128 | 128 | 128 | 128 |
| Number of Attention Head | 16 | 16 | 16 | 16 | 16 | 16 | 16 | 4 | 4 | 4 | 4 | 4 |
| Attention Score Dropout | 0.1 | 0.1 | 0.1 | 0.1 | 0.1 | 0.1 | 0.1 | 0.0 | 0.1 | 0.0 | 0.1 | 0.0 |
| Token Embedding Dropout | 0.1 | 0.1 | 0.1 | 0.1 | 0.0 | 0.0 | 0.0 | 0.1 | 0.0 | 0.1 | 0.1 | 0.1 |
| Feedforward Dropout | 0.2 | 0.2 | 0.2 | 0.2 | 0.2 | 0.2 | 0.2 | 0.2 | 0.2 | 0.2 | 0.2 | 0.2 |
| Normalization Layer Position | Post | Post | Post | Post | Post | Post | Post | Pre | Post | Post | Pre | Pre |
| Number of Layers | 8 | 8 | 8 | 8 | 4 | 4 | 4 | 6 | 4 | 5 | 8 | 6 |

dropout layer after multi-head-attention and feedforward network is called token embedding dropout, and a dropout layer after the feedforward network's dimension expansion linear layer is called feedforward dropout.

The classification head used in ModelNet40 dataset is a bit different: it uses our differentiable sampling to sample 512 tokens and then uses a maximum aggregation to aggregate information. All of the classification heads we used in our experiments except the one on ModelNet40 use a linear layer with softmax aggregation.

Table 9 shows architectural hyperparameters of experiments.

