# OpenReview forum: "SFT: Sampling-based Foundational Transformer"
_TMLR — Rejected by TMLR_

### Review · Reviewer_baJj · 2024-05-17

**Summary Of Contributions:**

The authors introduced a set of modifications to the transformer architectures that includes (1) a sampling method for self-attention that significantly reduces the computational complexity in terms of the number of tokens (2) a new non-linearity that leads to convex characteristics. The new non-linearity has some supporting theoretical results.

**Audience:**

Yes

**Claims And Evidence:**

Yes

**Requested Changes:**

I will use this section to ask questions, since I will also request many of the clarifications to be included in the paper as well.

1. Overall, I would like the authors to define all mathematical notations used precisely. This may feel pedantic, but I am having a lot of trouble understanding key components of this work at the moment. I will list the ones that most directly confused me:

1a. Is the $\*$ symbol used for matrix multiplication? In Line 15 of Algorithm 2, is $\*$ here also matrix multiplication?

1b. What is the onehot function? What is lower case $x_0$ in equation 2, maybe a row of $X$? Where does $P$ show up in Algorithm 1?

1c. In an equation between (2) and (3), you defined Gumbel(0,1) as a function, but then in line 6 of Algorithm 1, you are adding a Gumbel(0,1)? Does this mean a sample from the Gumbel distribution? What is the dimension of this Gumbel(0,1) object in Algorithm 1?

1d. In line 7 of Algorithm 1, what does it mean to sample from $S$? What is this object $\pi$? What is the operation $K[\pi]$?

1e. Similarly, you did not define $F_1, F_2$, Prob, Score. What are these functions?

1f. Right before equation 6, you said "we can derive the trivial sampling with replace procedure by repeatedly applying Equation 3 or Equation 5." However, do you mean here drawing a sample from the distribution defined by equation 3 or 5?

1g. You did not define the ImportanceScore or the Concat function in equation 6 and 7.

2. Secondly, for Theorem 4.2 and 4.3, what is the architecture being analyzed here? Is this for one single layer of the attention block?

3. In Theorem 4.2, can you intuitively summarize what "certain combinations of weights" mean? I don't think referring to the Appendix is helpful here for the reader.

4. Theorem 4.3, when you use the big $\Theta$ notation, what is the variable going to infinity? Is it just the number of tokens $n$?

I will stop here for now, as there is already a lot to be clarified. I will return to the discussion and attempt to evaluate the work again once these questions are addressed.

**Strengths And Weaknesses:**

I have quite a bit of clarification I would like to resolve with the authors before returning to summarizing the strength and weaknesses. In particular, since I'm not an expert on the training of the transformers, I will mainly focus on evaluating the theoretical component of this work for its correctness and clarity.

---

> ### Author Response · Authors · 2024-05-25
> **Response to Reviewer baJj (1)**
>
> Thank you for pointing these out. We will provide a list of used notations as revision and clarify some here:
>
> tl;dr: Most of the confusion is caused by our misuse of notations on matrix-multiplication/hadamard-elementwise-product, sample operation/expression of random variables, and the usage of notations from programming language (tensor slice).
>
> - **[1a]:**  Is the $\*$ symbol used for matrix multiplication? In Line 15 of Algorithm 2, is $\*$ here also matrix multiplication?
>   - Yes it is. This is our typo in writing; the $*$ in Line 15 is supposed to Hadamard product/Elementwise product. We will fix the mentioned and related typos by using solely $\odot$ for Hadamard product.
> - **[1b]:** What is the onehot function? What is lower case $x_0$ in equation 2, maybe a row of $X$? Where does $P$ show up in Algorithm 1?
>   - One-hot function defined in equation (2) maps from the index of a token in the input representation to a vector in $\mathbb{R}^n$. This output vector contains binary values including 0 and 1 where the value at the $i^{th}$ position is 1 if $i$ is chosen index and 0 otherwise.
>   - The lower case $x_0$ indicates the index in the input representation of the token with highest importance score. Likewise, $x_1$ indicates the index of the second most important, $x_2$ indicates the third, and so on.
>   - $P$ is equivalent to the sampling procedure in the Algorithm 1. In other words, $PX=X[\pi]$ where $\pi$ is defined in Algorithm 1.
> - **[1c]:** In an equation between (2) and (3), you defined Gumbel(0,1) as a function, but then in line 6 of Algorithm 1, you are adding a Gumbel(0,1)? Does this mean a sample from the Gumbel distribution? What is the dimension of this Gumbel(0,1) object in Algorithm 1?
>   - In the equation between (2) and (3), the Gumbel(0,1) is the PDF of the distribution itself. We will clarify that by adding in the distribution Gumbel(0,1), we meant to add a random sample from the distribution. We consider adding random samples drawn from Gumbel(0,1) to be elementwise, which is for each element in the added tensor, we sample a real number from Gumbel(0,1) and add to that element. Specifically in Algorithm 1, the dimension of the sampled tensor should be identical to the dimension of $(\mathbf{X} * W_S + B_S)$, which is a tensor of importance score with shape (n_tokens, 1). In practice, there's a dimension called batch size but since each batch is processed in parallel and independently, we opt to not include batch dimension.
> - **[1d]:** In line 7 of Algorithm 1, what does it mean to sample from $S$? What is this object $\pi$? What is the operation $K[\pi]$?
>   - $S$ is a tensor. Line 7 of Algorithm 1 decribes this procedure: samples a random tensor from Gumbel(0,1) probability distribution, then added the sampled tensor to the importance score, saved as $S$.
>   - Sample from $S$ describes the procedure in Algorithm 2.
>   - $\pi$ is a list containing the indices of tokens ordered by their importance score.
>   - The operation $K[\pi]$ means slicing the tokens defined by their indices in $\pi$ and stacking (concatenating) them.
> - **[1e]:** Similarly, you did not define $F_1, F_2$, Prob, Score. What are these functions?
>   - $F_1, F_2$ are activation functions. These are function typically map from $\mathbb{R}\to\mathbb{R}$. Functions such as $ReLU(x)=\max(0,x)$ or $Softplus(x)=\log(1+\exp(x))$ are activation functions. If the input of an activation function is a multidimensional tensor, then the mapping operates element-wise and outputs a tensor of the same size. This is later relevant on Line-4, page-9, where we construct our relative positional encoding with both multiplicative and bias terms.
>   - Prob are the functions that behave like Softmax function, which is a core part of transformer architectures and classification problems. It takes vectors of logits and converts them to probability vectors.
>   - Score functions are functions that convert two $\mathbb{R}^d$ vectors into a scalar $R$. In vanilla transformer architecture, scaled-dot product is used to compute attention score.
>   - Our alternatives of Softmax function and scaled dot product are shown in Section 4.3.
>   - The details of Prob and Score functions are later elaborated in Section 4.3, as function f and function g, respectively. We chose the words Prob and Score in the Overview part of method section to provide intuition and we are sorry that this instead brought confusion. We will find ways to revise our text to address this problem.

---

> ### Author Response · Authors · 2024-05-25
> **Response to Reviewer baJj (2)**
>
> - **[1f+g]:** Right before equation 6, you said "we can derive the trivial sampling with replace procedure by repeatedly applying Equation 3 or Equation 5." However, do you mean here drawing a sample from the distribution defined by equation 3 or 5? You did not define the ImportanceScore or the Concat function in equation 6 and 7.
>   - Short answer:
>     - No, the line implies an algorithm that directly applies that stochastic function multiple times in parallel since sample with replacement is independent. It is certainly not the case for sampling without replacement.
>     - The ImportanceScore function computes the "importance score" of each token (each row of a matrix of shape $\mathbb{R}^{n\times d}$) and outputs a vector of length $n$ containing real positive values. The calculation is done token-wise via equation (4). Verbally, it's our expression of a learnable linear layer.
>
> - Long answer (you can skip this, we add for clarity):
>     - The reason we did not write in detail the sampling with replacement procedure is that it is not used in our work.
>     - To properly answer this question, we believe a brief on the reparameterization trick is necessary. The technique is used in Gumbel-Softmax [1].
>     - Choosing is not a differentiable function. Their idea of making choosing differentiable is using another differentiable function's gradients to optimize, and that function should be similar to the non-differentiable function. Before this work, at the foundation of deep learning, the sigmoid function resembles the non-differentiable step function in Boolean networks. These two ideas are similar.
>     - This means that in the forward pass, it returns the results of a non-differentiable choosing function but in the backward pass, it computes the gradient of the differentiable function. For example, softmax is the soft version of argmax function, in forward pass, argmax is used; in backward pass, the gradient of softmax is used to update parameters.
>     - There's a slight difference: Gumbel-Softmax uses a variable $\tau$ to control the "temperature". When $\tau\rightarrow0^{+}$, the modified function becomes more similar to argmax function but is harder to optimize.
>     - Concat is short for concatenation; e.g. Concat([4, 3, 5], [2, 3]) = [4, 3, 5, 2, 3]; this works for multi-dimensional array / tensor as well.
>     - Equations 3 and 5 are stochastic functions. $g_i$ are random real number sampled independently from Gumbel(0, 1) distribution. Equation 3 or 5 are used to do sampling.
>     - Equations 3 or 5 are computed $k$ times and the sampled tokens are concatenated into a tensor.
>     - If using Gumbel-Softmax reparameterization trick, we have a discrete sample as softmax is replaced by the argmax in forward pass.
>
> [1]: Eric Jang, Shixiang Gu, Ben Poole. Categorical Reparameterization with Gumbel-Softmax. ICLR 2017.

---

> ### Author Response · Authors · 2024-05-25
> **Response to Reviewer baJj**
>
> - **[2]:** Secondly, for Theorem 4.2 and 4.3, what is the architecture being analyzed here? Is this for one single layer of the attention block?
>   - In Theorem 4.2 and Theorem 4.3, the architecture used here is indeed a single block of transformer with the attention formula replacing dot product and softmax with max out and probability ReLU respectively. However, we removed the sampling component in these theorems because sampling (slicing matrices) does not affect pseudoconvexity. In terms of gradient magnitude, the expectation of $||Sa/W_K||_F^2$ can become $\Theta(k)$ when sampling top-k most important tokens, i.e. k tokens of highest importance scores.
>
> - **[3]:** In Theorem 4.2, can you intuitively summarize what "certain combinations of weights" mean? I don't think referring to the Appendix is helpful here for the reader.
>   - In Theorem 4.2, "pseudoconvex w.r.t certain combinations of weights" means that the function is not pseudoconvex completely. Instead, by setting some variables constant, it is pseudoconvex with respect to the remaining variables. This can be much interpreted as componentwise pseudoconvex but the components are multiple real variables instead of only one.
> - **[4]:** Theorem 4.3, when you use the big $\Theta$ notation, what is the variable going to infinity? Is it just the number of tokens $n$?
>   - In Theorem 4.3, it is correct that the gradient magnitude complexity is calculated with respect to only $n$, the number of tokens in the input representation.

---

> > ### Comment · Reviewer_baJj · 2024-06-04
> > **Response**
> >
> > Thank you for the response.
> >
> > Let me start by commenting (partly for the action editor) that while the large number of undefined notations is confusing, the main issue is that I found it difficult (and unpleasant) to evaluate the mathematical content of this paper. I don't think it should be the reviewers' job to help fix basic notational issues, so I hope the authors can avoid these issues for future submissions.
> >
> > That being said, it's completely possible to fix these issues and convince me to raise my evaluation of the paper. I would simply request the authors to update the manuscript with all the changes (from your response) highlighted in a new colour, so that it is easy to tell apart the changes. From what I see righ tnow,
> >
> > Here are several follow up questions.
> >
> > 1b. So what is the purpose of defining multiple notations for sampling? You have defined $\text{Sam}(X), S(X), PX$, and $X[\pi]$. Some of which I have not seen reused at this point.
> >
> > 1b. How would you distinguish the $x_i$ notation in equation 2 with the $x_i$ notation in equation 5? It seems like in the latter, the input is the unsorted tokens. Additionally, you defined the function $s(x,z,g)$ in equation 5, but how does it extend to inputs $X, \mathcal{S}$?
> >
> > 1b+g. In equation 7, you used $\text{Sam}(X,k)$, is this consistent with the earlier notation of $\Sam{X}(k)$, since you are suggesting Concat is defined as horizontal which outputs a one dimensional object?
> >
> > 1d. Algorithm 2 takes two inputs $X$ and $k$, not $S$. Is this what you mean by $\text{Sample}(S)$? When you use the notation $\pi \sim \text{Sample}(S)$, does this imply that the output of $\text{Sample}(S)$ is a probability distribution over the collection of all possible one-hot matrices $P$?
> >
> > 3. Can you clarify exactly what the function that is "multiple-component-wise" convex? And did you use this pseudo-convexity property in establishing Theorem 4.3?
> >
> > 4. I have two follow up questions. (4a) Since the derivative with respect to the value weights is scaling with the number of tokens and the query is constant, does that mean there is no regime where both of these gradients can be non-vanishing/exploding at the same time?
> >
> > (4b) I thought exploding and vanishing gradients were mostly a problem with respect to the depth of the network based on my reading of existing work. Do the authors know of any work studying whether or not gradients vanish/explode with respect to the number of tokens?

---

> ### Author Response · Authors · 2024-06-11
> **Response to Reviewer baJj**
>
> We deeply appreciate your constructive and meticulous feedback, our work is still between revisions and may be partly confusing. Here, we have collated our answers for your questions:
>
> - So what is the purpose of defining multiple notations for sampling? You have defined $\text{Sam}(X), S(X), PX$, and $X[\pi]$. Some of which I have not seen reused at this point.
>   - We clarify that \text{Sam}(X), S(X) represent sampling as a function. PX is used to define the sampling procedure with mathematical formulation. And, X[pi] is the standard matrix slicing in pytorch code, which indicates that the rows of X is selected and ordered by the indices in the ordered set pi \subset {1,2,...,n}
> - How would you distinguish the $x_i$ notation in equation 2 with the $x_i$ notation in equation 5? It seems like in the latter, the input is the unsorted tokens. Additionally, you defined the function $s(x,z,g)$ in equation 5, but how does it extend to inputs $X, \mathcal{S}$?
>   - The x_i notation in equation 2 should be the order ranked by importance score of the i^th token in the input representation.
>   - In equation 5, x is a matrix and z is a vector. This means that x_i is the i^th token and z_i is the i^th tokens’ importance score. These definitions align with the input X and \mathcal{S}
> - In equation 7, you used $\text{Sam}(X,k)$, is this consistent with the earlier notation of $\Sam{X}(k)$, since you are suggesting Concat is defined as horizontal which outputs a one dimensional object?
>   - We have revised Algorithm 2 for better understanding. The concat function stack the tokens (rows) sorted by importance score and stack them vertically. The output of the Concat function should be a k\times d matrix.
>   - We redefined S in Algorithm 2. The vector S should contain all n importance scores of n tokens in the input.
> - Algorithm 2 takes two inputs X and k, not S. Is this what you mean by Sample(S)? When you use the notation $\pi\sim\text{Sample}(S)$, does this mean the output of $Sample(S)$ is a probability distribution over the collection of all one hot matrices $P$?
>   - We realize that there might be some misunderstanding here. I think you mean Algorithm 1 when refering to the output of $Sample(S)$. For this, we suggest you take a look at the pseudocode we added in out latest revision in Fig 1 and Fig 2.
>
> - Can you clarify exactly what the function that is "multiple-component-wise" convex? And did you use this pseudo-convexity property in establishing Theorem 4.3?
>   - (Pseudo)convex w.r.t a certain set of variables means that: There is a choice of a set of variables such that if the other variables are constant then the function is (pseudo)convex. For example, if f(a,b,c) is (pseudo)convex  with respect to (a,b) then if c is treated as constant $\overline{c}$ then $f(a,b,\overline{c})$ is (pseudo)convex. In our case, the output of the SFT block is pseudoconvex with respect to the three set of weights (W_Q, b_Q, W_K, b_K) , (W_V,b_V) and (W_C,b_C)
>   - We did not use pseudoconvexity in Theorem 4.3.
> - I have two follow up questions. (4a) Since the derivative with respect to the value weights is scaling with the number of tokens and the query is constant, does that mean there is no regime where both of these gradients can be non-vanishing/exploding at the same time? I thought exploding and vanishing gradients were mostly a problem with respect to the depth of the network based on my reading of existing work. Do the authors know of any work studying whether or not gradients vanish/explode with respect to the number of tokens?
>   - We have adjusted the meaning of Theorem 4.3. Our main goal is to make an analysis on the magnitude of the gradient to avoid flat and sharp points during training. And for that, with complexity bounded expectation for the gradient norm, we wish to ensure stable training. We realize that gradient vanishing/exploding is not derivable via single layer gradient analysis. Additionally, whatever property that holds for W_Q is also applicable to W_K since their roles in the formula are symmetric.
>   - The vanilla formula attention has proven to be difficult to analyze in terms of gradient [https://arxiv.org/abs/1703.00091] [https://openreview.net/forum?id=SkMuPjRcKQ]. The difficulty comes essentially from the softmax formulation, which is hard to calculate the expectation. Some works have over-simplified things by treating the attention matrix as an uniform matrix [https://arxiv.org/pdf/2206.03126 ] (Assumption 3.1) but such an assumption is not realistic in both theory and practice. By changing the attention formulation from softmax to relu and maxout, we have successfully proved the boundedness in the expectation of the gradient norms.
>
> I hope these responses suit you well. Thank you again for the review.

---

### Review · Reviewer_gsf2 · 2024-05-26

**Summary Of Contributions:**

The paper addresses two major challenges in the use of transformers for sequence processing: the high computational cost of self-attention and the complexity of training transformers. To tackle these issues, the authors propose two innovative mechanisms: a neural-guided down-sampling method for self-attention and a new attention non-linearity that is both linear-scaled and convex. The evaluation of the proposed method was based on point clouds, graphs, and long-range sequences. The source code is provided to help the objective evaluation.

**Audience:**

Yes

**Broader Impact Concerns:**

Not applicable.

**Claims And Evidence:**

Yes

**Requested Changes:**

Add additional experiments about the performance of fine-tining the long context reasoning ability from large language models.

**Strengths And Weaknesses:**

Strengths:
-  The paper is well-written. The organization of the paper is clear. Most technique details are easy to follow.
- There are concrete theoretical analyses of the proposed method.

Weakness:
- My main concern with the paper is the lack of discussion on applying the proposed method in pertaining or fine-tining the long context reasoning ability from large language models, which seems to be the most matched scenario for the proposed method. The lack of the technique discussion and the corresponding evaluation leaves some logical flaws in the motivation of this paper.

---

> ### Author Response · Authors · 2024-06-11
> **Rebuttal**
>
> Dear Reviewer gsf2,
>
> Thank you for reading our manuscript. We are sorry for deliver this rebuttal late.
>
> - We have make certain update on motivation and what we are aiming in our manuscript.
>
> - The experiment to train our model as a LLM is very costly, which renders pretraining LLM financially impossible. As it is a new architecture with lots of modification, fine-tune is actually impossible. Therefore, as our best effort to address your concerns, we are currently training our model on WikiText103 and include the result in the final manuscript.
>
> - There are two shortcomings:
>   - Our model is not implemented as a compiled single kernel, nor applied FlashAttention for better throughput. Therefore, we will use softmax-dot formulation, which is equivalent to only use our sampler stack on FlashAttention. This has certain limitation but at least it shows the practicality of our sampler. This would save cost of training.
>   - Sequential nature:
>     - Causal LM has sequential nature.
>     - We can modify our model to be causal by including not attend to the sampled token has larger index/position.
>     - This has the problem on text generation as the sampled 256-token from 1024 first tokens is different to the sampled 256-token from (1024+1) tokens. This means it is hard to reuse KV, which is the reason of using causal language model (KV-cache).
>     - This rules out a number of possibilities, such as, put the sampler on top of LLAMA (which break causality) and fine-tune.
>     - A base model without causality (BERT-like) can be used to fine-tune, where the base BERT weight is frozen (cold/blue/snowflake) and the sampler's weight is trained (hot/red/fire) like the adapter literature. We had to read the adapter literature as we were not familiar with it. This requires significant engineering work and we are considering it. We don't believe this can be delivered in time but we will try.
>     - We are training our model as a bidirectional next word generator and will report the perplexity of our model in final version.
>   - Training to get perplexity is not sufficient to evaluate the long context reasoning.
>
> - We hope you understand certain limitations in experimental capacity and our model's adaptability in reusing other model's weight.

---

### Review · Reviewer_Jqs5 · 2024-05-28

**Summary Of Contributions:**

Transformers attention efficiency is the long standing problem which current paper also tries to solve. Authors propose three new modifications to make attention linear (or n logn, n - sequence length) 1) modification of attention to sub-sample keys (which gives sparse attention) via first sorting the keys based on the scoring function and then taking the top-k: here authors proposes scheme on this selection so that gradient through the sampling can be computed to properly propagate through the top keys; 2) non-linearity instead of softmax - in combination with the sparsity provides pseudo-convexity property for the new attention block which should improve the training stability (ease of hyper-parameters search) and speed up convergence; 3) new relative positional embedding scheme, which accepts now graphs, point clouds and long sequences for the transformer model. Authors show efficiency of the proposed model by proving pseudo-convexity property and bound on the gradient norms, and further demonstrate results with empirical analysis on point clouds, long-range-graph benchmarks and long-range arena benchmarks. Empirical results show that model performs in the top of other models (but do not outperform the SOTA in each domain).

**Audience:**

Yes

**Broader Impact Concerns:**

Overall the work is theoretical and empirical regarding the basic modeling in the machine learning. As other models it can lead to any outcomes in the future, but with the current formulation it is more about core ML. Authors listed limitations, I would add that broader testing of the system is needed across domains and applications.

**Claims And Evidence:**

No

**Requested Changes:**

> the paper writing needs a lot of improvements: a lot of inconsistency between naming; math notation is very unclear and inconsistent - it is really hard to read and parse all theoretical formulations (took me a lot of time, though in most cases I got the idea / what authors mean but formality is really ambiguous).

- be consistent with usage: "foundation" or "foundational", camel case or lower case for "Transformers", "relational" positional embedding is not used in the literature - it should be "relative" - be consistent on usage too, "hyperparameters" or "hyper-parameters", spaces between words and references, references with citep / citet depending on the case (right now it is inconsistent)
- typos:
  - "an empirical analysis of the ability of our transformer" - ability on what?
  - "messing passing" -> "message passing"
  - "in the Appendix" -> "in Appendix"
  - "one of the first transformer" -> "one of the first transformers"
  - Linear(X) - is not clear as then it is map of vector to vector, and here input is tensor.
  -  page 4 "PE" is not introduced I believe
  - page 9 "optimzations" -> "optimizations"
  - page 11 "attention from between"?
- introduction - why there is no references to vision and speech e.g. as applications for transformers to sequences?
- "sparsity patters" - I think variants of linear attention are not necessary sparsity patterns, it is just the linearization of the attention operation, so would be nice to discuss this too.
- Figure 1: randomization is not clear from the figure, the color for rows doesn't correspond to the color of "randomization" word. Also $C$ is incorrectly shown as it is based on the $V$ but not on $X$.
- "over-centralized attention" - what is it? I don't know this definition.
- Sec 4.1. reference to Section .. - missed section index.
- Math notation is very unclear:
  - tensors, vectors, scalars are not consistently marked with bolt / not bolt. Indexing is not clear for the vectors / tensors.
  - $X = (x_1, .., x_n)\in \mathbb{R}^{n\times d}$ -> $X = (x_1, .., x_n)^T\in \mathbb{R}^{n\times d}$
  - "Sam", "S" - why different notation in Sec 4.2? also $nm$, $km$ - add \times.
  - Eq 2 is unclear, as not clear what onehot exactly doing and what is $x_i$ here as it is not bolt (I know what eq 2 is doing, but formally is it very poorly notated).
  - Usage of $*$ between matrices / tensors for matrix multiplication is bad notation, it is a convolution then op, not matmul.
  - Algorithm 1 row 7 - Sample is not defined
  - Eq 3 where are vectors, scalars? what is $\tau$? do you set temperature to infinity to be able to sample only one vector instead of linear combination?
  - Eq 4 - how do you select only one vector and not linear combination?
  - Eq. 8 - proof / discussion how you got it?
  - Algorithm 2 row 3 - how do you concat vector and $\mathbf{S}$ which is scalar? All notations between scalars and vectors are messed up here. Please either use pytorch pseudo-code or do correct math notations / ops here. It is really better to have pytorch pseudo-code, it took me a while to parse what is written. $\mathcal{S}_1[\mathbf{S}]$ notation is also very unclear. Row 16 - do we return linear combination of vectors? otherwise how do we get that only top vector is selected from the softplus op? Why do we need also Gumbel in row 2?
  - Eq 9 - again notation on tensors, vectors, index of the tensors are poor.
  - What is $max(\mathbf{X_i}, \mathbf{Y_j})$? is it max per coordinate?
  - What is $\mathbf{X_r}$? learnable? sinusoidal?
- "It is well-known that transformers are hard to optimize" - the reference here is about post-LayerNorm models and very deep models, however in general pre-LayerNorm and smaller models are fine to optimize. In era of LLMs we know now (let me know if you need references) that bigger models (not deeper) it is also hard and a lot of instabilities occurs. I would reference here to more works, which mention these later issues.

> Pseudo-convex property in Def 4.1 is wrongly given thus the proof on pseudo-convexity is questionable.

It should be at the end of the definition $f(x_2) \geq f(x_1)$, while in the paper it is written $f(x_1) \geq f(x_2)$. The latter is wrong. The same definition is used in the proof in the Appendix, thus the correctness of the proof is under question now as incorrect definition is used.

> There is absence of many ablations:

As I specified, it will be stronger paper if proper ablations are done to show why we need every component or what better properties we have for the model as soon as it doesn't outperform SOTA models.

**Other questions**
-  why do you consider post-layernorm models?
- what sample rate is used in all reported empirical results?
- why in the end model doesn't outperform prior works? what is then the point of this new architecture?
- "The higher convergence rate of our model ..." - where and how do you see higher convergence rate?
- Figure 5: what is the difference between "softmax dot" and "vanilla"? is it positional embedding?
- "vanilla transformer does not support relational information" - what do you mean exactly? I can add relpos into transformer, moreover it is learning it anyway.
- Table 6: what are numbers for 100% sampling rate? what if you use relpos for vanilla transformer?
- page 38: "learnable linear combination of sinusoid1D" - what is that? could you give exact formulation what positional embedding is used?
- page 39: "we introduce dropout" - can you confirm if the same is used for other model? also why your model overfits and you need this extra dropout? could it be that your model is performing better only because of this dropout?

**Note**: I did a pass over Appendix with the proof and only didn't do deep reading for Appendix A3 as found the error (possible?) in the definition and thus proof for the earlier part.

**Strengths And Weaknesses:**

**Strengths**
- interesting idea/trick on making k-top sampling without replacement differentiable (not sure though how novel this is, but the idea/technique is definitely worth sharing / discussing)
- construction of the transformer block to be pseudo-convex, as pseudo-convex property is nice property to have
- theoretical and empirical results, which include benchmarking across several domains

**Weaknesses**

See requested changes for all details for every weakness.

- the paper writing needs a lot of improvements: a lot of inconsistency between naming; math notation is very unclear and inconsistent - it is really hard to read and parse all theoretical formulations (took me a lot of time, though in most cases I got the idea / what authors mean but formality is really ambiguous).
- I think authors didn't demonstrate how new formulation of attention prevents from transformers training difficulty. Just saying the model pseudo-convex now - I think is not enough. Moreover, it is not clear if the whole model is pseudo-convex as I can have some other blocks before or after transformers blocks. Also pseudo-convexity restricts the model a lot and we see in empirical results that model cannot outperform some SOTA models, then I don't get the point of introducing so restrictive model if in the end we restrict the expressiveness of it.
- Pseudo-convex property in Def 4.1 is wrongly given thus the proof on pseudo-convexity is questionable.
- There is absence of many ablations:
  - ablate introduced relative positional embedding - why do we need new one? what happens if we use RoPE or relative sinusoidal one or relative learnable one? even for the introduced relative positional embedding I didn't get is it learnable of sinusoidal? Why do we need such modification on the positional embedding?
  - what happens if we don't use $C$? in that case we loose pseudo-convexity (if I assume that all derivations in the paper are correct) but how about performance and model capacity then?
  - no analysis on the robustness to hyper-parameters and comparison between deep / non-deep models, pre-layernorm / pos-layernorm - as they are the main source of instabilities and issues of transformer training.
  - what is performance / speed if no sampling of keys happening (assuming that we have full attention)? do we outperform standard attention?
  - comparison with standard attention on regular tasks with no keys subsampling to see that the idea on the transformer modification itself works (here comparison should be fair in the sense of same positional embedding e.g.)
- If I missed it, then it should be clearly stated / expanded - why do we need introduced modifications compared to prior works? I got about new variant of differentiable sampling, but why we need pseudo-convexity (as we restrict the family too much then) and new variant of relative positional embedding (as no comparison or deeper analysis are given) are not clear.

---

> ### Author Response · Authors · 2024-06-11
> **Rebuttal (1/n)**
>
> Dear Reviewer Jqs5,
>
> Thank you for reading our work thoroughly (took lots of your time to comprehend what is going on in the manuscript). We are sorry to deliver the rebuttal late.
>
> - be consistent with usage: "foundation" or "foundational", camel case or lower case for "Transformers", "relational" positional embedding is not used in the literature - it should be "relative" - be consistent on usage too, "hyperparameters" or "hyper-parameters", spaces between words and references, references with citep / citet depending on the case (right now it is inconsistent)
>   - We have fixed the mentioned errors in the revision.
> - typos:
>   - "an empirical analysis of the ability of our transformer" - ability on what?
>     - We have included "on the performance and efficiency on popular point cloud, graph, sequence benchmark" of our model.
>   - "messing passing" -> "message passing"
>     - We have this error in the revision
>   - "in the Appendix" -> "in Appendix"
>     - We have fixed this error in the revision.
>   - "one of the first transformer" -> "one of the first transformers"
>     - We have fixed this error in the revision.
>   - Linear(X) - is not clear as then it is map of vector to vector, and here input is tensor.
>     - We takes the last two dimensions of tensor to multiply with the weight matrix, which we believe to be a common practice, reference: https://pytorch.org/docs/stable/generated/torch.nn.functional.linear.html
>   - page 4 "PE" is not introduced I believe
>     - We are not quite understand this. To parse our understanding, we think you refer to the three Positional Encodings in GraphGPS paper. According to what we understand of the GraphGPS work, they categorized graph positional encodings to three kinds and combine into one model.
>   - page 9 "optimzations" -> "optimizations"
>     - We fixed this typo in our revision.
>   - page 11 "attention from between"?
>     - We fixed this typo to "attention between..." in our revision.
> - "sparsity patters" - I think variants of linear attention are not necessary sparsity patterns, it is just the linearization of the attention operation, so would be nice to discuss this too.
>   - We have introduced this work in our original related work, we believe. " Subsequent work
> like Linear Transformer (Katharopoulos et al., 2020) redesigns the similarity kernel previously as a SoftMax of dot-product score (or any other kernels) to one made of a linear probability cage." which is similar to yours "it is just the linearization of the attention operation".
> - Pseudo-convex property in Def 4.1 is wrongly given thus the proof on pseudo-convexity is questionable.
>   - We have fixed the pseudoconvex definition in both main text and appendix. We have double-checked and made sure that our proofs was correct as it was based on Lemma B.4 (in our latest revision) and not the definition.
> - If I missed it, then it should be clearly stated / expanded - why do we need introduced modifications compared to prior works? I got about new variant of differentiable sampling, but why we need pseudo-convexity (as we restrict the family too much then) and new variant of relative positional embedding (as no comparison or deeper analysis are given) are not clear.
>   - The vanilla formula attention has proven to be difficult to analyze in terms of gradient [https://arxiv.org/abs/1703.00091] [https://openreview.net/forum?id=SkMuPjRcKQ]. The difficulty comes essentially from the softmax formulation, which is hard to calculate the expectation. Some works have over-simplified things by treating the attention matrix as an uniform matrix [https://arxiv.org/pdf/2206.03126 ] (Assumption 3.1) but such an assumption is not realistic in both theory and practice. By changing the attention formulation from softmax to relu and maxout, we have successfully proved the boundedness in the expectation of the gradient norms.
>   - We clarify that learnable sinusoidal is probably not new. We introduce leakiness as we discover some cases that transformer with relative positional encoding cannot learn. To demonstrate, we will give an example. There is a problem to distinguish two graph where node value can be {a, b} and edge value can be {0, 1}. Transformer cannot distinguish any two graph have: same number of nodes, different edge connection when naively use adjacency matrix as attention bias because softmax sums to 1. Since our transformer works on point cloud, graph, and sequence; we need to fix the mentioned problem. The analysis on this is given on Figure 8.

---

> ### Author Response · Authors · 2024-06-11
> **Rebuttal 2/n**
>
> - Figure 1: randomization is not clear from the figure, the color for rows doesn't correspond to the color of "randomization" word. Also $C$ is incorrectly shown as it is based on the $V$ but not on $X$.
>   - We have updated the figure with less transparent color and fixed the $C$ in our revision. This error is not easy to see (really thanks for pointing it out). The sampled vectors are different from the unsampled vectors (which is a linear combination of a "top-k" vector and noise).
> - "over-centralized attention"
>   - Over centralization is the phenomenon of the attention mechanism in which a small number of tokens dominates the attention score. This happens due to the softmax formulation which only highlight the top values [https://arxiv.org/abs/2308.16898]
> - Sec 4.1. reference to Section .. - missed section index.
>   - We have fixed this in our revision
> - Math notation is very unclear
>   - We are currently fixing this; in the meantime, we have made torch pseudocode for our sampling without replacement mechanism. We will make torch pseudocode for our entire method overview. For the overview algorithm, the updated figure should be able to show sufficient details on what going on. These pseudocode is introduced side-by-side with figures.
> - Eq 3 where are vectors, scalars? what is $\tau$? do you set temperature to infinity to be able to sample only one vector instead of linear combination?
>   - In the original Gumbel Softmax work [https://arxiv.org/abs/1611.01144], they effectively set tau to 0.0 at inference.
> - Eq 4 - how do you select only one vector and not linear combination?
>   - We use linear combination of the top component and the other component (we treat the other component as sampling noise). However, this can be used with Gumbel-Softmax reparameterization trick and the reason we do not use the trick is that it hurts convergence rate. We are leaning towards using regularization (to make it as “sampling at convergence”) at the latter training stage of the model to increase the magnitude of importance score; or to penalize stochasticity: (model(input, random_seed1) - model(input, random_seed2)) ^ 2 regularization, where our stochastic model is called twice.
>
> - Sec 4.1. reference to Section .. - missed section index.
>   - We have fixed this in our revision
> - Math notation is very unclear
>   - We are currently fixing this
>
> - "It is well-known that transformers are hard to optimize" - the reference here is about post-LayerNorm models and very deep models, however in general pre-LayerNorm and smaller models are fine to optimize. In era of LLMs we know now (let me know if you need references) that bigger models (not deeper) it is also hard and a lot of instabilities occurs. I would reference here to more works, which mention these later issues.
>   - Yea, we need some of references. We have seen works like ReZero initialization which assists help convergence rate.
>   - Like other work (MEGA: arxiv.org/pdf/2209.10655) We also consider both types of transformers. There’s a trade-off in pre-norm and post-norm: pre-norm suffers from representation collapse; therefore, we think it’s worth to consider both.

---

> ### Author Response · Authors · 2024-06-11
> **Rebuttal 3/n**
>
> - There is absence of many ablations
>   - We agree since we introduce many components in our model. Our ablation study so far have covered: the effect of sampling rate (both performance and efficiency) (Figure 7), computational cost of each submodule (Table 7), effect of inclusion of relative information (Figure 6), Similarity between Tokens (Figure 8), and performance gaps of relative information only (Table 5). It is hard to remove components and run on many experiment as we have certain constraint on resources.
>   - We cannot outperform SOTA since there's many data-modality specific inductive bias introduce in these models. We thought it was impossible to design inductive bias that works for every data modality.
>   - We will try to answer some ablation study related:
>      - ablate introduced relative positional embedding - why do we need new one? what happens if we use RoPE or relative sinusoidal one or relative learnable one? even for the introduced relative positional embedding I didn't get is it learnable of sinusoidal? Why do we need such modification on the positional embedding?
>        - It is not the new "learnable linear combination of sinusoidal embeddings" that make our model can accept point cloud, long range sequences, and graph. The learnable linear combination of sinusoidal embeddings only helps in sequential tasks, which is our attempt to introduce a kind of inductive bias that favor locality into our model. We did not delve deeply into this because we thought this was not significant in any sense and merely an engineering trick. However, we will provide intuitive explaination in this rebuttal:
>        - Sinusoid1D consists of sinusoid wave with different frequencies.
>        - Each position in the sequence receives different positional
>        - When applying the same linear transformation to positional embeddings of different positional information to different tokens, they match based on the same frequencies.
>        - This is our attempt to introduce inductive bias.
>      - what happens if we don't use $C$? in that case we loose pseudo-convexity (if I assume that all derivations in the paper are correct) but how about performance and model capacity then?
>           - The component $C$ is a positive valued vector added to the denominator during attention matrix calculation. This inclusion is motivated from the theoretical work at [https://arxiv.org/pdf/2205.13401]. The authors here suggested two conditions for universal approximation of transformers without absolute positional encoding, i.e. X -> X+E. The model capacity will decrease to current transformer RPE's capacity if removing $C$. Check our Figure 3 and 8.
>           -  (Pseudo)convex w.r.t a certain set of variables mean that: There is a choice of a set of variables such that if the other variables are constant then the function is (pseudo)convex. For example, if f(a,b,c) is (pseudo)convex  with respect to (a,b) then if c is treated as constant $\overline{c}$ then $f(a,b,\overline{c})$ is (pseudo)convex. In our case, the output of the SFT block is pseudonconvex with respect to the three set of weights (W_Q, b_Q, W_K, b_K) , (W_V,b_V) and (W_C,b_C).
>      - what is performance / speed if no sampling of keys happening (assuming that we have full attention)? do we outperform standard attention?
>        - Relative information is important for point cloud, as every point cloud model use it. It is also important for sequential model and graph. Performance wise, we would outperform a pure transformer model, given it is a stack of nn.TransformerEncoderLayer. Speed wise, no, we would be slower than standard attention, where the threshold is 25%. There are many factor to be accounted, one of such is that nn.TransformerEncoderLayer uses FlashAttention.
>      - comparison with standard attention on regular tasks with no keys subsampling to see that the idea on the transformer modification itself works (here comparison should be fair in the sense of same positional embedding e.g.)
>        - We did try to compare on one simple and cheap to train task under Figure 6 (between orange and blue lines)

---

> > ### Author Response · Authors · 2024-06-11
> > **Rebuttal 4/n**
> >
> > Other questions:
> > - why do you consider post-layernorm models?
> >   - Representation collapse in pre-layernorm model; somehow it hurts the performance of our point cloud models.
> > - what sample rate is used in all reported empirical results?
> >   - Less than or equal to 25% as it is the threshold of "faster empirically". The exact sampling rate can be found by dividing the number of sampled tokens (which we provided in Appendix) by the number of tokens.
> > why in the end model doesn't outperform prior works? what is then the point of this new architecture?
> > - "The higher convergence rate of our model ..." - where and how do you see higher convergence rate?
> >   - The orange line is consistently above the blue line in Figure 6 is what made us believe.
> > - Figure 5: what is the difference between "softmax dot" and "vanilla"? is it positional embedding?
> > "vanilla transformer does not support relational information" - what do you mean exactly? I can add relpos into transformer, moreover it is learning it anyway.
> >   - It is now Figure 6 in our revised manuscript. Yes, we meant the relative positional encoding. It is true that any absolute positional encoding can learn a relative one. We add our relpos into our transformer with prob function is SoftMax and score function is scaled dot product. If adding relying solely on relative positional encoding using vanilla transformer, some of the tasks could have failed (e.g. all token embeddings are the same and relative positional encoding is Euclidean Distance Matrix for SO-3 Point Cloud modeling). Reason: attention map might be different but it sums to one. The token embeddings is the same -> every token embeddings after transformed through a self attention layer is also the same. This simple reasoning illustrates the limitation of relpos transformer in transferring relative positional encoding information to token embeddings. More information can be found in this interesting work [https://arxiv.org/abs/2205.13401].
> > - Table 6: what are numbers for 100% sampling rate? what if you use relpos for vanilla transformer?
> >   - It should be slower due to relpos. If using our relpos for vanilla transformer, it should make it slower; if using the built-in attention-bias, it should be neglectiably slower. Our answer is based on our Table 7.
> > - page 38: "learnable linear combination of sinusoid1D" - what is that? could you give exact formulation what positional embedding is used?
> >   - We will add the details. For now, we will provide informal formulation.
> > - page 39: "we introduce dropout" - can you confirm if the same is used for other model? also why your model overfits and you need this extra dropout? could it be that your model is performing better only because of this dropout?
> >   - We include it as the benchmark datasets are small. Other model has much stronger inductive bias that can apply to these datasets. For our comparison of our model against vanilla-like transformer (softmax+dot), the extra dropout is used. Every component between softplus-maxout (our) and softmax-dot (vanilla) is the same (PE, dropout, ...) except these two components: prob and score function. We have discovered recently that the modification is discovered first in this work: https://aclanthology.org/2021.naacl-main.302/, Figure 1b. We will cite their paper in revision.

---

> > > ### Comment · Reviewer_Jqs5 · 2024-06-24
> > > **Revised paper review 1/N**
> > >
> > > Dear authors,
> > >
> > > Sorry for the delay, it took some time to go through the paper and proofs again. Please find below my additional comments on your answers and revised paper.
> > >
> > > > We takes the last two dimensions of tensor to multiply with the weight matrix, which we believe to be a common practice, reference: https://pytorch.org/docs/stable/generated/torch.nn.functional.linear.html
> > >
> > > Yep, I got what you are doing, but mathematically how you write and define it text was incorrect.
> > >
> > > > page 4 "PE" is not introduced I believe
> > >
> > > I mean the notation of the abbreviation for PE. Introduce that Positional Embedding is PE.
> > >
> > > > "sparsity patters"
> > >
> > > Mainly I have the issue with the term you use. As e.g. linear attention is not notated as "sparsity patters" in the prior works. So citing is good, but calling all as "sparsity patters" is not correct in my opinion.
> > >
> > > > We have fixed the pseudoconvex definition in both main text and appendix. We have double-checked and made sure that our proofs was correct as it was based on Lemma B.4 (in our latest revision) and not the definition.
> > >
> > > Main issue I had is the proof in ThrB6: Last inequality in the proof is wrong; it should be -> $P(W_{C_2}) \geq P(W_{C_1})$ to satisfy the pseudo-convexity definition. By rechecking it I see that all derivations seem to be correct except this last inequality, which gives the wrong definition of the pseudo-convexity. So please just fix the type of inequality, and then it is ok.
> > >
> > > > The vanilla formula attention has proven to be difficult to analyze in terms of gradient…
> > >
> > > This should be clearly discussed / pointed out in the text. But then if in practice we use another form of attention - what is the connection of theory for maxout with softmax attention especially in the conditions that softmax still outperforms maxout? What particular properties we inherit from proving something for maxout that extend or tell us about softmax one? I think a strong connection between why we do this theoretical analysis is needed. On its own it is ok, but why we need it - not clear for me. There should be a reason why we want to do this theory.
> > >
> > > > We clarify that learnable sinusoidal is probably not new. We introduce leakiness as we discover some cases that transformer with relative positional encoding cannot learn.
> > >
> > > I think the scope then is mixed for the paper, which is not good for readability, and also then empirical results are not proving necessity of that positional embedding. Either you have separate paper for the new positional embedding to cover other cases, or you do in-depth analysis of the new positional embedding extending paper into two pieces - attention analysis and then positional embedding. And here motivation for new positional embedding should be clearly shown with the empirical results.
> > >
> > > > Over centralization is the phenomenon of the attention mechanism in which a small number of tokens dominates the attention score. This happens due to the softmax formulation which only highlight the top values [https://arxiv.org/abs/2308.16898]
> > >
> > > Thanks for pointing, this is similar to what I had in mind while reading. However, I don’t see this term “over-centralized attention” in the reference you provided. So my complaint is to the terms you introduce in the paper, while there is either another term in prior works or no such term at all. So either notate it or use another wording.
> > >
> > > About pytorch pseudo-code vs math notation: I believe as the paper about theoretical analysis all notations and algorithms should be clearly stated in math formulation as this is what is used for analysis for different bounds / operations. However yep, it is a good idea to have pseudo-code for people who would like to use your attention right away. So in my opinion pseudo-code should be in the appendix with reference in the main body, while clear math should be done in the main body.
> > >
> > > >  In the original Gumbel Softmax work [https://arxiv.org/abs/1611.01144], they effectively set tau to 0.0 at inference.
> > >
> > > Does it mean that you set it to 0 too? But then how do you sample exactly?
> > >
> > > > It is not the new "learnable linear combination of sinusoidal embeddings" that make our model can accept point cloud, long range sequences, and graph. The learnable linear combination of sinusoidal embeddings only helps in sequential tasks, which is our attempt to introduce a kind of inductive bias that favor locality into our model. We did not delve deeply into this because we thought this was not significant in any sense and merely an engineering trick. However, we will provide intuitive explanation in this rebuttal
> > >
> > > As you are saying in the intro “with our new relative positional encoding scheme which works (adequately) for many kinds of data modalities.” - this means that you claim this as a part of contribution. This leads me to question why we need it and how good it is, what is the difference exactly with all prior works. In that case either down the tone on that or do a proper empirical justification.

---

> > > > ### Comment · Reviewer_Jqs5 · 2024-06-24
> > > > **Revised paper review 2/N**
> > > >
> > > > > We did try to compare on one simple and cheap to train task under Figure 6 (between orange and blue lines)
> > > >
> > > > Are orange and blue the same settings except the attention formulation?
> > > >
> > > > > Representation collapse in pre-layernorm model; somehow it hurts the performance of our point cloud models.
> > > >
> > > > Yep, this is what we observe in other domains too, but then the post-layer norm is harder to train. So if in your new attention in the end you show that post-LayerNorm is stable (as gradients are bounded) you should highlight this more and have a bit discussion why you go with post-LayerNorm - maybe in the end we could use your new attention for LLMs / foundation models training as then post-LayerNorm is more expressive and now stable?
> > > >
> > > > > The orange line is consistently above the blue line in Figure 6 is what made us believe.
> > > >
> > > > This kind of contradicts your own words before that model cannot outperform softmax classic formulation and that you cannot beat SOTA. But now you say it is actually better. I think more careful empirical formulation what you exactly observe is needed. And it is ok from my point of view to say that you are doing parity comparison between methods with not SOTA model (but close) w/o extra engineering tricks people developed to get SOTA.
> > > >
> > > > > It is now Figure 6 in our revised manuscript. Yes, we meant the relative positional encoding. ….
> > > >
> > > > Thanks for the pointer, and yep agree on the points you mentioned. Main confusion is again the way you write it and define things. Vanilla transformer is used to denote softmax dot product attention with absolute sinpos embedding. All other variations should be explicitly mentioned with pointing out what positional embedding is used.
> > > >
> > > >
> > > > **New comments**:
> > > > - In general I think the writing needs more work even after revision.
> > > > - Abstract “without any modifications.” - you actually do modification of vanilla transformer, i think this should be changed in text e.g. “general modification applicable to several data modalities. About “For efficiency on the training convergence rate and to ease the pain of meticulous hyper-parameter tuning” - I still think there is not enough empirical evidence provided in the main text to justify this. Maybe good to look into analysis for hyper-parameters pain in https://arxiv.org/abs/2309.14322.
> > > > - I feel that paper still overloaded: 1) theory 2) new model and positional embedding 3) stability and no pain for hyper-parameters tuning (I don’t think it is shown, broader number of experiments and analysis needed) 4) inference speed up (not clear what is model performance on the accuracy / other metrics is, as you can be worse with accuracy but faster - known trade-off) 5) applicability to many domains. Some work is needed to make it more structural - the paper with the revision is improved significantly but still it is overloaded and hard to follow all findings and reasoning under them.
> > > > - From introduction it seems relpos is the central thing - which is not and you contradict above all discussion where you say that relpos is not what you bring as novelty.
> > > > - “applicability on (almost) all data modalities and transformer variants” - what do you mean here under “transformer variants”?
> > > > - For Figure 1 I suggest using einops lib for the pytorch pseudo-code to have better readability for tensor dimensions and remove the block scheme or simplify it, maybe showing main differences with the vanilla transformer?
> > > > - “by repeatedly applying Equation 6 or Equation 8” -> “by repeatedly applying Equation 5 or Equation 8”. So eq 5 and 8 are the same, right? you always take argmax, the only difference in your sampling is how you compute gradient as the straight-through operator?
> > > > - English and text overall for the revised text needs proofread - hard to read sometimes (though I got the meaning), punctuation is weird sometimes.
> > > > - Th B6: please fix reference to Algo 1, Formula before (16) doesn’t have $C_4$. Last inequality in the proof is wrong; it should be $P(W_{C_2}) \geq P(W_{C_1})$ to satisfy the pseudo-convexity definition.

---

> > > > > ### Author Response · Authors · 2024-06-30
> > > > > **Reponse to review on revised paper (1/N)**
> > > > >
> > > > > Dear reviewer Jqs5,
> > > > >
> > > > > Really thank you for your constructive response.
> > > > >
> > > > > tl;dr: We are currently scanning for the last errors and many reduction in claims.
> > > > >
> > > > > > "Yep, this is what we observe in other domains too, but then the post-layer norm is harder to train. So if in your new attention in the end you show that post-LayerNorm is stable (as gradients are bounded) you should highlight this more and have a bit discussion why you go with post-LayerNorm - maybe in the end we could use your new attention for LLMs / foundation models training as then post-LayerNorm is more expressive and now stable?"
> > > > > "
> > > > > - We tried to measure in our SO-3 experiment, from cosine similarity of 1.0 at the first layer dropping down to as low as 0.4. We see that RPE can clearly solve representational collapse trivially (if having leaky component). We believe rank injection is much different to other parametrization, where they aim to mitigate and we aim to ensure it never happens. It is the Figure 8 for trained network and Figure 3 for untrained network.
> > > > >
> > > > > > "This should be clearly discussed / pointed out in the text. But then if in practice we use another form of attention - what is the connection of theory for maxout with softmax attention especially in the conditions that softmax still outperforms maxout? What particular properties we inherit from proving something for maxout that extend or tell us about softmax one? I think a strong connection between why we do this theoretical analysis is needed. On its own it is ok, but why we need it - not clear for me. There should be a reason why we want to do this theory."
> > > > > - There are certain connections can be derived from this way of thinking. One of such is a similar function that resembles dot product similarity, which is also linear scaled as makes a transformer layer pseudoconvex: abs(q+k) - abs(q-k). This is zero-centered, same sign -> higher attention score, different sign -> lower attention score. We have it in the code, but choose not to report it because it is slow to compute. You can see the plot of the function on WolframAlpha for the similarity: https://www.wolframalpha.com/input?i=plot+abs%28q%2Bk%29+-+abs%28q-k%29
> > > > >
> > > > > > "I think the scope then is mixed for the paper, which is not good for readability, and also then empirical results are not proving necessity of that positional embedding. Either you have separate paper for the new positional embedding to cover other cases, or you do in-depth analysis of the new positional embedding extending paper into two pieces - attention analysis and then positional embedding. And here motivation for new positional embedding should be clearly shown with the empirical results."
> > > > > - We will reduce the claims accordingly; it's just we find that by replacing SoftMax -> Leaky SoftMax, where the Leaky component should scale with the number of key vectors (different to the Leaky in "attention sink/add_zero_attn" in PyTorch; does not scale with the number of tokens), increase what a RPE-based Transformer can model. This is evidenced in the SO-3 experiment.
> > > > >
> > > > > > "Does it mean that you set it to 0 too? But then how do you sample exactly?"
> > > > > - Reparameterization trick, the gradient function/backprop is computed different to forward function/forward prop. In other words, it is (hardmax(score, dim=-1) - F.softmax(score, dim=-1)).detach() + F.softmax(score, dim=-1). It works but much slower speed of convergence, therefore, we use tau=1.0 for two passes. We have clarified this in our revised manuscript.
> > > > > - For sampling, it is more like 0+/epsilon/1e-9. Think it is like softmax([-inf, -inf, -inf, 0, -inf]) = [0, 0, 0, 1, 0]; gradient is detached.
> > > > >
> > > > > > "Are orange and blue the same settings except the attention formulation?"
> > > > > - Yes, for fairness in comparison.
> > > > >
> > > > > > "This kind of contradicts your own words before that model cannot outperform softmax classic formulation and that you cannot beat SOTA. But now you say it is actually better. I think more careful empirical formulation what you exactly observe is needed. And it is ok from my point of view to say that you are doing parity comparison between methods with not SOTA model (but close) w/o extra engineering tricks people developed to get SOTA."
> > > > > - SOTA (specific enhancement like KNN tree, voxelization for point cloud, adding GCN layer, subgraph sampling for graphs, exponential coefficients for sequence) outperforms our model, for obvious reasons. Our model outperforms vanilla transformer. While our model should be compared against vanilla transformer only, we feel it is crucial to include other specialized models to see where our model is. A similar research of new attention nonlinearity conducts their experiment comparing against softmax-dot, for example: FourierFormer[https://openreview.net/pdf?id=PRd7VG_ki_]

---

> > > > > > ### Author Response · Authors · 2024-06-30
> > > > > > **Reponse to review on revised paper (2/N)**
> > > > > >
> > > > > > > "I feel that paper still overloaded: 1) theory 2) new model and positional embedding 3) stability and no pain for hyper-parameters tuning (I don’t think it is shown, broader number of experiments and analysis needed) 4) inference speed up (not clear what is model performance on the accuracy / other metrics is, as you can be worse with accuracy but faster - known trade-off) 5) applicability to many domains. Some work is needed to make it more structural - the paper with the revision is improved significantly but still it is overloaded and hard to follow all findings and reasoning under them."
> > > > > > - The reason of it is overloaded deeply rooted on the progression of the research. Sampling was hard to converge, often, the gradient just explodes (we was trying to use sorting network for top-k-sampling). There were two different approaches: to make transformer easier to be optimized or to make sampling easier to be optimized. Making transformer easier to be optimized worked for 5 layers of Sampled-based Transformer with sorting network converges (it is not good enough but still), while the other direction led to the current sampling method. We included both in this manuscript.
> > > > > > - We believe in larger dataset, it is fated to perform worse than full attention with the same number of hidden dimension and layers; we do not know if it can scale with larger number of layer enabled by reduction in computational cost.
> > > > > >
> > > > > > > "Abstract “without any modifications.” - you actually do modification of vanilla transformer, i think this should be changed in text e.g. “general modification applicable to several data modalities. About “For efficiency on the training convergence rate and to ease the pain of meticulous hyper-parameter tuning” - I still think there is not enough empirical evidence provided in the main text to justify this. Maybe good to look into analysis for hyper-parameters pain in"
> > > > > > - We will reduce our claim in hyperparameter tuning thing. We do not have the infrastructure to train 100M+ parameter model for this research.
> > > > > >
> > > > > > > "In general I think the writing needs more work even after revision."
> > > > > > - We will try to think another perspective to restructure the manuscript. Currently, we believe we have listed all the steps needed to construct our model.
> > > > > >
> > > > > > > "For Figure 1 I suggest using einops lib for the pytorch pseudo-code to have better readability for tensor dimensions and remove the block scheme or simplify it, maybe showing main differences with the vanilla transformer?"
> > > > > > - We will try einops. Things would get more tedious if we try to sample each attention head differently tbh; around 120-ish lines of PyTorch.
> > > > > >
> > > > > > > "“by repeatedly applying Equation 6 or Equation 8” -> “by repeatedly applying Equation 5 or Equation 8”. So eq 5 and 8 are the same, right?"
> > > > > > - You are right, we should put these two equations into one.
> > > > > >
> > > > > > > "English and text overall for the revised text needs proofread - hard to read sometimes (though I got the meaning), punctuation is weird sometimes."
> > > > > > - We will proofread the thing again.
> > > > > >
> > > > > > > "Th B6: please fix reference to Algo 1,..."
> > > > > > - We will fix it.
> > > > > >
> > > > > > > "“applicability on (almost) all data modalities and transformer variants” - what do you mean here under “transformer variants”?"
> > > > > > - Our sampling method can be easily applied into Flash Attention (query attended to key/value *flashly*/fast), BigBird (replacing the random attention to guided), Reformer (enhance Reformer via addition of Global Attention), Point Transformer V1/2/3 (Makes attention faster), Vision Transformer (allowing more image patches, the sampled tokens can be thought as learnable patch patterns), ... Apart from Flash Attention, combining with any other mentioned methods removes our model's applicability towards other data modalities.
> > > > > >
> > > > > > Thanks again for reviewing.

---

> > > > > > > ### Comment · Reviewer_Jqs5 · 2024-07-01
> > > > > > > **Reply**
> > > > > > >
> > > > > > > Dear authors,
> > > > > > >
> > > > > > > Thanks for the reply and additional clarifications.

---

### Author Response · Authors · 2024-06-11
**Summary of our latest revision**

We acknowledge that our manuscript has consistencies in notations as reviewers have pointed out. Since there are still a lot to be fixed, we believe making an official comment on what has been fixed in this revision is needed. We will fix our manuscript in the next days (under a week).

- We have elaborate the motivation of our work in a more detailed way in Abstract and Introduction sections.
- We have provided "Notations" section in Appendix and use $AB$ as matrix multiplication of matrix $A$ and matrix $B$ instead of $*$.
- We have updated the figures and provide a PyTorch styled pseudocode for the simplified sampling procedure (the whole procedure with relative information is hard to provide / illustrate in the figure because the code was much longer). We believe the simplified version should provide reader sufficient information on our ideas and the new figure of the layer should show better what is going on in the model.
- We have fixed many typos and the pseudoconvex definition.

Thank you all reviewers for pointing these inconsistencies out. We very much appreciate it.

---

### Decision · Action_Editor_Teu4 · 2024-07-17

**Recommendation:** Reject

**Comment:**

The paper presents novel ideas on improving the efficiency of self-attention which is an important challenge for Transformers, along with a new positional encoding, and generalizations to handle other data modalities. While all the reviewers (and I) agree that the paper has good potential, there were several issues raised that were not sufficiently addressed in the revisions. In particular, the issues were
- readability of technical content: this has improved but still needs work
- over-claiming and lack of support for certain claims: the jump from theory to empirical is not clearly motivated or backed
- overall structure of the paper: there seems to be a lot packed into the paper, but not easy to parse
- weak empirical results: lack of ablations for the main claims, and unclear performance gains

Given these, I recommend reject for the current submission and encourage the authors to take the reviewer feedback, incorporate it into an updated version, and re-submit. With improved readability, and adjustment of claims, I believe the paper would be good for TMLR.

**Audience:**

Yes, the paper is appropriate for TMLR.

**Claims And Evidence:**

Not entirely. As the reviewers point out, several claims in the paper (some have been adjusted in the revisions) do not have enough supporting evidence.

_Reviewer Jqs5's concerns (post-rebuttal): "The motivation to do that theory and switch to that transformer model is not clear. Either we have practically better model for which we can prove something (which is not the case) or we have then motivation from theory on changing the arch to show some good properties (stable training e.g.) and then making connection to real applications with vanilla transformer (if models in the end worse in practice). I think better writing is needed for both cases to motivate better and show connections."_

**Resubmission Of Major Revision:**

The authors may consider submitting a major revision at a later time.